# The contribution of asymptomatic SARS-CoV-2 infections to transmission on the Diamond Princess cruise ship

Jon C Emery, Timothy W Russell, Yang Liu, Joel Hellewell, Carl AB Pearson, CMMID COVID-19 Working Group, Gwenan M Knight, Rosalind M Eggo, Adam J Kucharski, Sebastian Funk, Stefan Flasche, Rein MGJ Houben*

Centre for Mathematical Modelling of Infectious Diseases, Department of Infectious Disease Epidemiology, Faculty of Epidemiology and Population Health, London School of Hygiene and Tropical Medicine, London, United Kingdom

**Abstract** A key unknown for SARS-CoV-2 is how asymptomatic infections contribute to transmission. We used a transmission model with asymptomatic and presymptomatic states, calibrated to data on disease onset and test frequency from the Diamond Princess cruise ship outbreak, to quantify the contribution of asymptomatic infections to transmission. The model estimated that 74% (70–78%, 95% posterior interval) of infections proceeded asymptomatically. Despite intense testing, 53% (51–56%) of infections remained undetected, most of them asymptomatic. Asymptomatic individuals were the source for 69% (20–85%) of all infections. The data did not allow identification of the infectiousness of asymptomatic infections, however low ranges (0–25%) required a net reproduction number for individuals progressing through presymptomatic and symptomatic stages of at least 15. Asymptomatic SARS-CoV-2 infections may contribute substantially to transmission. Control measures, and models projecting their potential impact, need to look beyond the symptomatic cases if they are to understand and address ongoing transmission.

*For correspondence:
rein.houben@lshtm.ac.uk

Group author details:
CMMID COVID-19 Working Group See page 11

Competing interests: The authors declare that no competing interests exist.

## Introduction

The ongoing COVID-19 pandemic has spread rapidly across the globe, and the number of individuals infected with SARS-CoV-2 outstrips the number of reported cases (*Wang et al., 2020*; *Golding et al., 2020*). One key reason for this may be that a substantial proportion of cases proceed asymptomatically, that is, they either do not experience or are not aware of symptoms throughout their infection but may still transmit to others. In this sense, asymptomatic infections differ from presymptomatic infections, which describes the portion of the incubation period before symptoms develop during which onward transmission is possible.

While pre- and asymptomatic individuals do not directly contribute to morbidity or mortality in an outbreak, they can contribute to ongoing transmission, as has been shown for COVID-19 (*Rothe et al., 2020*; *Chen et al., 2020*; *Ganyani et al., 2020*) and other diseases (*Dean et al., 2016*; *Slater et al., 2019*; *Esmail et al., 2018*). In particular, purely symptom-based interventions (e.g., self-isolation upon onset of disease) will not interrupt transmission from asymptomatic individuals and hence may be insufficient for outbreak control if a substantial proportion of transmission originates from pre- and asymptomatic infections (*Chen et al., 2020*).

An estimate of the proportion of infections that progress to symptomatic disease, also known as the case-to-infection ratio, provides an indicator of what proportion of infections will remain undetected by symptom-based case detection (*Salomon and COVID-19 Statistics, Policy modeling, and Epidemiology Collective, 2020*). Evidence so far have suggested that the proportion of SARS-CoV-

2 infections that proceed asymptomatically is likely non-trivial (*Li et al., 2020*; *Liu et al., 2020a*; *He et al., 2020*; *Mizumoto et al., 2020*; *Lavezzo et al., 2020*; *Bendavid et al., 2020*), although empirical data are often difficult to interpret due to opportunistic sampling frames (*Nishiura et al., 2020*) combined with low (*Fontanet et al., 2020*) and imbalanced participation from individuals with and without symptoms (*Gudbjartsson et al., 2020*). While it is likely that transmission from asymptomatic individuals can occur, (*Bai et al., 2020*) quantitative estimates are effectively absent. Improved understanding of the relative infectiousness of asymptomatic SARS-CoV-2 infection, and its contribution to overall transmission will greatly improve the ability to estimate the impact of intervention strategies (*Salomon and COVID-19 Statistics, Policy modeling, and Epidemiology Collective, 2020*). What is known is that in the presence of active case-finding, presymptomatic infections and symptomatic cases contribute almost equally to overall transmission, as both modelling and empirical studies have shown (*Liu et al., 2020a*; *He et al., 2020*).

Documented outbreaks in a closed population with extensive testing of individuals regardless of symptoms provide unique opportunities for improved insights into the dynamics of an infection, as knowledge of the denominator and true proportion infected are crucial, yet often unavailable in other datasets. Here, we use data from the well-documented outbreak on the Diamond Princess cruise ship to capture the mechanics of COVID-19 in a transmission model with explicit asymptomatic and presymptomatic states to infer estimates for the proportion, infectiousness and contribution to transmission of asymptomatic infections. Available data included the date of symptom onset for symptomatic disease for passengers and crew, the number of symptom-agnostic tests administered each day, and the date of positive tests for asymptomatic and presymptomatic individuals (*Mizumoto et al., 2020*; *Nishiura, 2020*; *NIID, 2020*).

## Results

### Model calibration

The model reflected the data well (*Figure 1*), including the differently timed peaks for confirmed symptomatic cases for crew (*Figure 1A*) and passengers (*Figure 1B*). In addition, the model matched the expected impact of quarantine of passengers on transmission from February 4th as illustrated by the drop in reproductive number (*Figure 1E*), followed by a later drop in transmission after February 10th, which was driven by a change in contact pattern in crew. See *Figure 1—figure supplements 1–2* for full calibration outputs.

### Asymptomatic infections

We estimated that 74% of infections proceeded asymptomatically (70–78%, 95% Posterior Interval (PI)) (see *Figure 2A*). The strong identifiability of this parameter is driven by the relative proportions of individuals testing positive with and without symptoms, combined with the time-delay between symptom-based and symptom-agnostic testing. As a result, our model estimated that in total 1304 (1,198–1,416) individuals were infected, representing 35% (32–38%) of the initial total population on the Diamond Princess. Over half of these infections had not been detected at disembarkation on February 21 st (53%, 51–56%) consisting of infected individuals who had recovered and became test negative before they were tested (37%, 34–40%), were yet to be tested (15%, 13–16%), or had recently been exposed and were not yet detectable at that point (1%, 1–3%). Nearly two-thirds of pre- and asymptomatic infections (67%, 66–68%) and 8% (6–9%) of symptomatic infections went undetected up until disembarkation (*Figure 2C*).

In contrast to the strong identifiability of the proportion of infections that were asymptomatic, the model was unable to identify the relative infectiousness of asymptomatic infections from the data, that is, a uniform prior was effectively returned (see *Figure 2B*). This is because the relative infectiousness of asymptomatic infections was degenerate with the overall contact rate, meaning the data were consistent with either relatively frequent contact with less infectious individuals or relatively infrequent contact with more infectious individuals (see *Figure 1—figure supplement 1*). Despite this, the estimated proportion of transmission due to asymptomatic infections is 69%, with a wide confidence interval (20–85%) and an interquartile range of 56–76% (*Figure 2D*). The reason this estimate is not effectively 0–100%, as might be expected by the unidentifiable relative infectiousness, is the combination of the strongly identified, relatively high proportion of infections that are

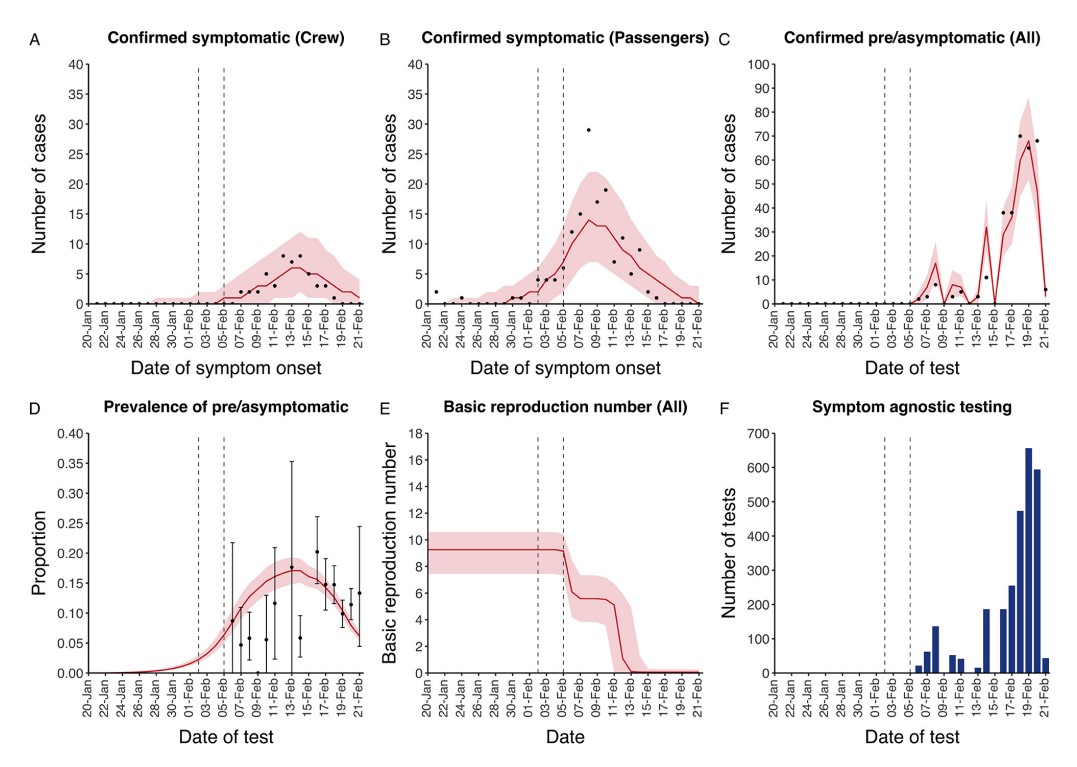

**Figure 1.** Data from the Diamond Princess and model calibration. Figure shows data from the Diamond Princess (points (A-D) and bars (F)) and results from model calibration. Red lines = median, shading = 95% posterior plus observational interval (A-C) and 95% posterior interval only (D-E). Two vertical lines show the date of the first confirmed diagnosis (left) and the start of quarantine measures (right). (A-B) show confirmed symptomatic cases among crew (A) and passengers (B) with a reported date of onset; (C) shows confirmed pre- or asymptomatic individuals by test date; (D) shows the prevalence of pre/asymptomatic individuals by test date. Points and error bars show point estimates and 95% confidence intervals; (E) shows the basic reproduction number over time for the ship as a whole, reflecting the drop in contact rates (F) shows the number of tests administered irrespective of symptoms, by test date.

The online version of this article includes the following source data and figure supplement(s) for figure 1:

**Source data 1.** Marginal posterior parameter values from model calibration.
**Figure supplement 1.** Parameter correlation plot from model calibration.
**Figure supplement 2.** Parameter trace plot from model calibration.

asymptomatic and the non-linear relationship between the relative infectiousness of asymptomatics and their contribution to transmission, which quickly saturates to its maximal value (see *Figure 2—figure supplement 1*). The result is that only a modest relative infectiousness is required to produce a non-trivial contribution to transmission. The relative infectiousness of presymptomatics was also unidentifiable, however, in all scenarios the remaining transmission was equally distributed between the presymptomatic (14%, 1–44%) and symptomatic (17%, 11–42%) individuals. *Figure 3* shows the instantaneous proportion of transmission from symptomatic (A), presymptomatic (B) and asymptomatic (C) individuals over the course of the epidemic.

Because of the non-identifiability of the relative infectiousness of asymptomatic infections we investigated marginal posterior estimates (*Table 1*). We find that low relative infectiousness of asymptomatic infections (0–25% compared to symptomatic individuals) would need to be compensated by a net reproduction number for individuals during their presymptomatic and symptomatic phase of 15.5–29.1.

## Sensitivity analyses

Without an asymptomatic state the model was unable to reconstruct the dynamics of the outbreak (*Appendix 2—figures 1–3*, Deviance Information Criterion (DIC) = 974 vs 329 for the primary

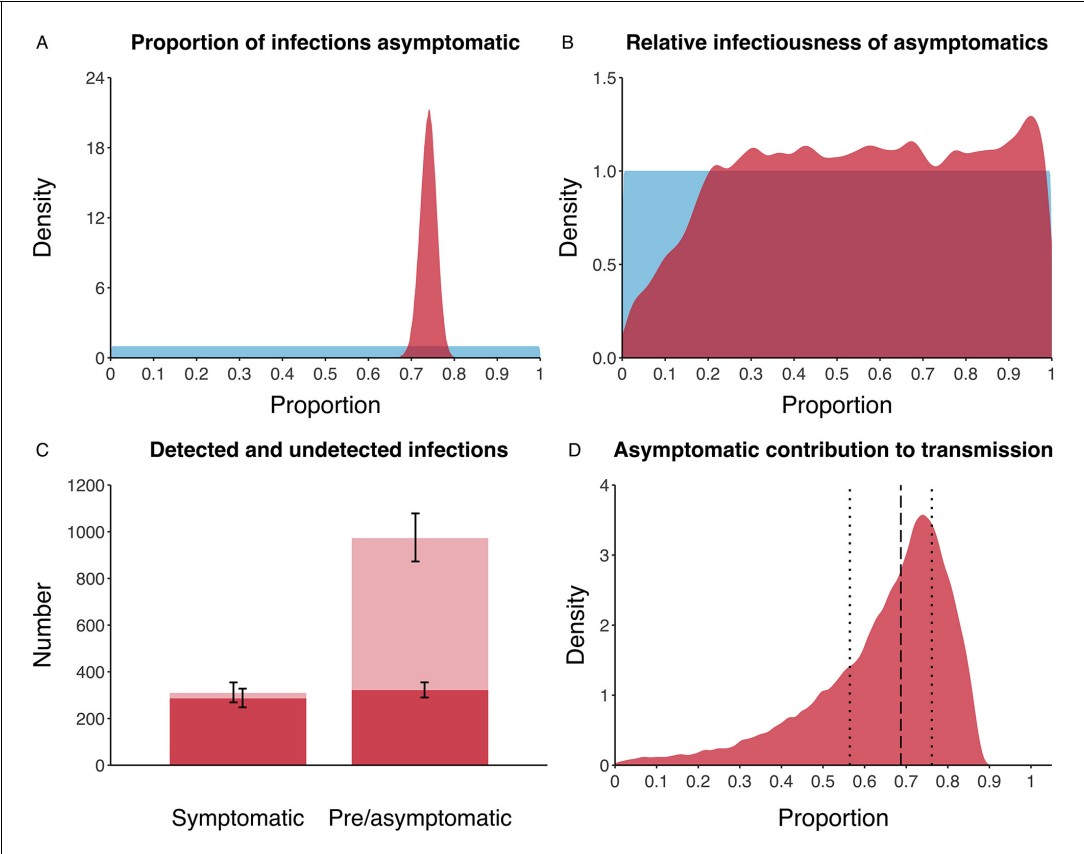

**Figure 2.** Proportion of infections that are asymptomatic and their contribution to transmission. (**A**) Prior (blue) and posterior (red) probability distribution for the proportion progressing to asymptomatic infections. (**B**) Prior (blue) and posterior (red) probability distribution for the relative infectiousness of asymptomatic infections. (**C**) Number of pre- and asymptomatic infections and symptomatic cases detected (dark red) and not detected (light red) during the outbreak. Error bars indicate 95% posterior intervals. (**D**) Posterior probability distribution for proportion of transmission that is from asymptomatic individuals. Dashed and dotted lines show median and interquartile range, respectively.

The online version of this article includes the following figure supplement(s) for figure 2:

**Figure supplement 1.** Non-linear correlation between relative infectiousness of asymptomatics and their contribution to transmission.

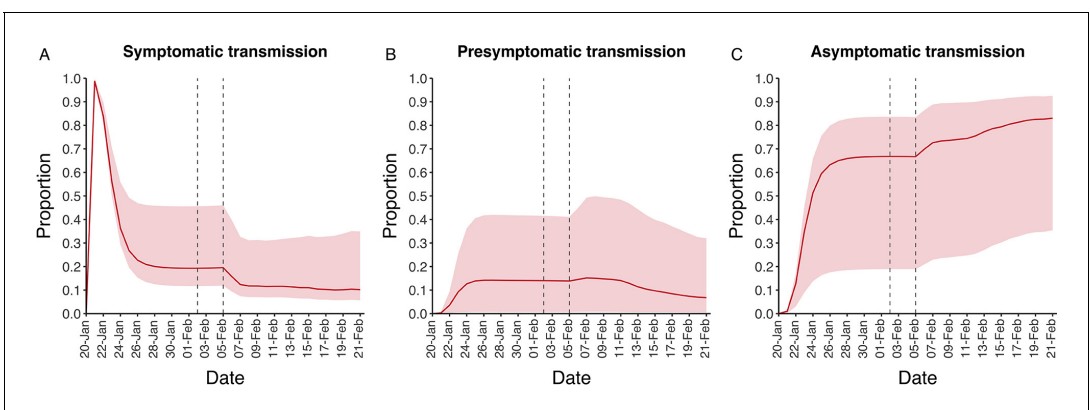

**Figure 3.** Instantaneous proportion of transmission from symptomatic, presymptomatic and asymptomatic individuals. Instantaneous proportion of transmission from symptomatic (**A**), presymptomatic (**B**) and asymptomatic (**C**) individuals over the course of the epidemic, following the introduction of a single symptomatic individual on 20th Jan. Red lines = median, shading = 95% posterior interval. Two vertical lines show the date of the first confirmed diagnosis (left) and the start of quarantine measures (right).

**Table 1.** Model outputs by relative infectiousness of asymptomatic individuals.
Relative infectiousness expressed as proportion compared to symptomatic individuals. All values are 95% posterior ranges from model scenarios. Net reproduction number represents the typical number of infections generated by a single infected individual during their presymptomatic and symptomatic stages.

| Range of relative infectiousness of asymptomatic individual | Model output | | |
| --- | --- | --- | --- |
| | Transmission from asymptomatic individuals (%) | Net reproduction number for presymptomatic passengers | Basic reproduction number |
| 0–1% | 0–3 | 22.7–29.1 | 6.7–7.6 |
| 1–25% | 7–58 | 15.5–25.5 | 7.0–8.8 |
| 25–50% | 44–75 | 11.1–17.6 | 8.0–9.6 |
| 50–75% | 60–82 | 8.7–13.6 | 8.7–10.2 |
| 75–99% | 68–86 | 7.2–11.4 | 9.3–10.8 |
| 99–100% | 72–87 | 6.7–10.2 | 9.5–10.9 |

Relative infectiousness expressed as proportion compared to symptomatic individuals. All values are 95% posterior ranges from model scenarios. Net reproduction number represents the typical number of infections generated by a single infected individual during their presymptomatic and symptomatic stages.

analysis). Moreover, adjusting the relative value for mixing between crew and passengers did not have a qualitative effect on the results (*Appendix 2—figures 4–11*).

Biased symptom-agnostic testing had a greater impact on the results. Those testing negative having a 50% higher probability of being tested compared to the primary analysis led to a corresponding greater proportion of infections that were asymptomatic (84%, 82–87%), total number infected (2,097, 1,914-2,292) and contribution of asymptomatics to transmission (78%, 36–91%) (see *Appendix 2—figures 12–15*). Conversely, those testing positive having a 50% higher probability of being tested compared to the primary analysis led to a corresponding smaller proportion of infections that were asymptomatic (66%, 61–71%), total number infected (1,051, 965-1,161) and contribution of asymptomatics to transmission (59%, 9–79%) (see *Appendix 2—figures 16–19*).

When we assumed a fixed age-specific ratio for the proportion of infections that progress asymptomatically, the model was able to fit the data, although the number of correlated parameters was high. Overall results were similar to the main analysis, with a proportion asymptomatic of 42% (41–44%) and 89% (85–91%) for passengers and crew, respectively. The proportion of all transmission from asymptomatics was 69% (IQR = 59–74%). Relative infectiousness was again unidentifiable. See *Appendix 2—figures 20–23* for details.

A longer latent period provided a poorer fit to the data (DIC = 361) (*Appendix 2—figures 24–27*). Adjusting the duration of the asymptomatic state to half or double the sum of the presymptomatic and symptomatic states made little qualitative difference to the results (*Appendix 2—figures 28–35*), although the shorter asymptomatic period was a marginally poorer fit to the data (DIC = 338). Finally, recalibrating the model assuming the 35 confirmed pre/asymptomatic cases where a test date was not available were allocated to the last feasible day (13th Feb) made no qualitative difference to our results (see *Appendix 2—figures 36–39*).

## Discussion

### Summary

We find that in this well-documented outbreak in a closed population, 74% (70–78%) of infections proceeded asymptomatically, equaling a 1:3.8 (1:3.3-1:4.4) case-to-infection ratio. The majority of asymptomatic infections remained undetected, but may have contributed substantially to ongoing transmission. While the relative infectiousness of asymptomatic infections could not be identified, low infectiousness (e.g. 0–25% compared to symptomatic individuals) would have required a very high net reproduction number for individuals during their presymptomatic and symptomatic stages of (15.5–29.1).

## Interpretation

Our results are strongly informed by data, which show that when extensive symptom-agnostic testing was ramped up, substantial numbers of pre- or asymptomatic infections were identified. Given the clear suppression of transmission through quarantining, as indicated by the drop in incident symptomatic disease, this finding is most likely explained by a large proportion of undetected asymptomatic individuals.

The model and data were unable to identify a value for the relative infectiousness, although we showed how different ranges for this key parameter required specific trade-offs, as reflected in the net reproduction number for infected individuals who will develop symptomatic disease. One can argue that a net reproduction number for presymptomatic passengers at the start of the outbreak of over 20 in this population, as required if asymptomatic individuals are effectively unable to transmit (range for relative infectiousness of 0–1%) is unlikely. Such high reproductive numbers are not usually seen, exceeding for example values found for norovirus outbreaks on cruise ships (*Gaythorpe et al., 2018*). While SARS-CoV-2 has been shown to survive on surfaces (*van Doremalen et al., 2020*), this does not seem to be the primary mode of transmission. In combination with growing evidence around viral load in asymptomatic infections and their involvement in transmission chains (*Lavezzo et al., 2020*), as well as anecdotal evidence about transmission from asymptomatic individuals (*Bai et al., 2020*; *Chau et al., 2020*) including in closed populations (*Arons et al., 2020*), it is reasonable to assume that asymptomatic infections play some role in ongoing SARS-CoV transmission. In our model, asymptomatic infections were responsible for more than half of all transmission in 83% of the scenarios compatible with the data.

It is important to note that our conclusion that asymptomatic infections may have contributed substantially to ongoing transmission is critically dependent on the setting. In this case symptomatic infections were quickly identified and removed from the ship before symptom-agnostic testing began, thereby leaving asymptomatic infections to dominate transmission. Such dominance should therefore not be interpreted as a constant of nature, but instead an important consideration in settings where prompt isolation of symptomatic infections is already in place but with little to no consideration for asymptomatic infections.

## Comparison to other studies

Our estimated proportion of asymptomatic infections in this outbreak is higher than previous studies, which relied on diagnosed cases only (*Mizumoto et al., 2020*). As we have shown, a substantial number of infections were not detected, which would explain some of the difference. Other empirical studies have found usually lower values, while some found similar ranges. While underestimation in other estimates due to low (*Fontanet et al., 2020*) and imbalanced participation from individuals with and without symptoms (*Gudbjartsson et al., 2020*) will be part of the explanation, there remains scope for unexplained variation from more complete samples (*Lavezzo et al., 2020*). In addition, it is possible that PCR-based testing has a lower sensitivity for asymptomatic individuals, which would further increase the proportion of infections that were asymptomatic (*Chau et al., 2020*).

A sensitivity analysis showed that our results were robust to age-specific probabilities of progressing to asymptomatic infections, as well as other assumptions made in the model, and driven by trends in the data.

Our estimated substantial contribution to transmission from asymptomatic infections confirms a hypothesis from Nishiura after analysing symptomatic cases occurring post-disembarkation (*Nishiura, 2020*). Our initial reproduction number of 9.3 (7.4–10.6) reflects the high transmission environment expected on cruise ships via increased contacts in a confined space, although lower than the value found in an earlier analysis by *Rocklöv et al., 2020*.

Our finding of similar contribution to transmission from presymptomatic and symptomatic individuals also matches findings by others (*Ganyani et al., 2020*; *Liu et al., 2020a*; *He et al., 2020*). In line with this, it is clear that having symptoms, or at least being aware of them, is not required in the transmission of SARS-CoV-2 (*Ganyani et al., 2020*; *Liu et al., 2020a*; *He et al., 2020*; *Kimball et al., 2020*; *Wei et al., 2020*). Although cough is often considered essential for transmission of respiratory infections (*Patterson and Wood, 2019*), work in tuberculosis, influenza and other coronaviruses has shown that while a cough may increase spread, it is not a requirement. Transmission from breathing,

talking and sneezing is also possible, as well as transmission from contaminated surfaces (*van Doremalen et al., 2020*; *Leung et al., 2020*; *Asadi et al., 2019*; *Williams et al., 2020*).

## Limitations

Additional data, in particular on the distribution of asymptomatic infections across crew and passengers, by age and shared quarantine environments would have benefited the model and potentially enabled us to estimate a range for the relative infectiousness of asymptomatic infections. A serological survey of the population, and the date and testing history of individuals who developed symptoms post-disembarkation, would also have likely informed more precise model estimates. In addition, better evidence on performance of the test used, and the associated likelihood of false-negative or false-positive results would help refine estimates. As more data become available, future model analyses of SARS-CoV-2 dynamics in closed populations should further inform the key questions we have looked to address here.

Whilst symptom-agnostic testing provided valuable insights into the pre- and asymptomatic states, such testing was not necessarily random, as was assumed in our primary analysis. Indeed, it is known that individuals were generally screened in reverse-age order (*NIID, 2020*). Sensitivity analyses considering biased testing still produced non-trivial results for the proportion of infections that were asymptomatic however.

Our model also assumed that the infectiousness of all transmissible states was constant over time. If instead symptomatic individuals are most infectious immediately after symptom onset (*He et al., 2020*), an estimate of their contribution to transmission would in principle increase. Given the likely heightened awareness of symptoms on board however, such an increase is likely to be marginal. Indeed, since our assumption of an average one-day, exponentially distributed delay between symptom onset and removal from the ship is likely to be an overestimate, a prompter removal distribution would at least in part offset such an increase.

A similar simplification was made by assuming that the probability of an individual progressing to either a presymptomatic or asymptomatic infection was independent of who infected whom. It is possible, however, that transmission from a symptomatic infection may be more likely to ultimately result in another symptomatic infection, owing to a higher infecting dose for example.

## Conclusion

Asymptomatic SARS-CoV-2 infections may contribute substantially to transmission. This is essential to consider for countries when assessing the potential effectiveness of ongoing control measures to contain COVID-19.

## Materials and methods

### Data

Data from the Diamond Princess outbreak have been widely reported. (*Mizumoto et al., 2020*; *Nishiura, 2020*; *NIID, 2020*) On January 20th, the Diamond Princess cruise ship departed from Yokohama on a tour of Southeast Asia. A passenger that disembarked on January 25th in Hong Kong subsequently tested positive for SARS-CoV-2 on February 1st, reporting the date of symptom onset as January 23rd.

After arriving back in Yokohama on February 3rd, all passengers and crew were screened for symptoms, and those screening positive were then tested. The ship began quarantine on February 5th with all passengers confined to their cabins and crew undertaking essential activities only. At the start of quarantine there were 3711 individuals on board (2666 passengers and 1045 crew) with a median age of 65 (45–75 interquartile range).

Testing capacity was limited until February 11th and before then the majority of individuals tested had reported symptoms, referred to here as 'symptom-based testing'. All individuals with a positive test at any stage were promptly removed from the ship and isolated. After February 11th, testing capacity increased and the testing of individuals irrespective of symptoms, referred to here as 'symptom-agnostic testing', was scaled up. In total, 314 symptomatic and 320 pre- or asymptomatic infections were reported before disembarkation was principally completed on February 21st.

We extracted the following data from *Mizumoto et al., 2020*; *Nishiura, 2020*; *NIID, 2020* (see *Figure 1*). Firstly, the number of symptomatic cases per day (i.e. those testing positive having reported symptoms) by date of symptom onset, separately for passengers and crew. The date of symptom onset was not available for 115 cases, which we accounted for in our model structure by assuming they were distributed over time proportional to those cases with a reported date of symptom onset (see *Appendix 1—table 1*). Secondly, we extracted the number of pre- or asymptomatic infections identified per day (i.e. individuals testing positive having not reported symptoms) by date of test. The test date was not available for 35 pre- or asymptomatic individuals between the February 6-14th, which we assumed were distributed over time proportional to the daily number of tests performed amongst individuals not reporting symptoms. No data were available on how many individuals that tested positive in the absence of symptoms became symptomatic after disembarkation. Finally, we extracted the number of tests performed per day amongst individuals not reporting symptoms (see *Appendix 1—table 2*).

## Model

We built a deterministic, compartmental model to capture transmission, disease development and the effect of interventions on board the Diamond Princess. Following exposure, after which an individual is assumed to test negative for SARS-CoV-2 for the duration of the latent phase (see *Table 2*), a proportion of individuals proceed asymptomatically with the remainder becoming presymptomatic. This proportion equates to a universal probability of becoming either presymptomatic or asymptomatic, independent of who infected whom. Individuals in the presymptomatic, asymptomatic or symptomatic state are assumed to test positive and have independent infectiousness, expressed relative to those with symptomatic disease.

Individuals with presymptomatic infection are either detected through symptom-agnostic testing before being removed from the ship, or develop symptomatic disease. Once symptomatic disease starts, individuals can either recover undetected on the ship or, following the start of quarantine on February 5th, be detected through symptom-based testing and removed from the ship with an average delay of one day following symptom onset. We allowed for individuals to test positive after their infectious period for an average of seven days (*Woelfel et al., 2020*). After this, we assume they would test negative.

Individuals with asymptomatic infections either recover undetected on the ship, or are detected by symptom-agnostic testing before being removed from the ship. See *Appendix 1—figure 1* for a diagram of the model.

Symptom-agnostic testing was assumed to have been random amongst those not reporting symptoms and no delay was introduced between testing and removal of those that tested positive from the ship. As such, the number of people that tested positive through symptom-agnostic testing before being removed from the ship per day was calculated using the number of tests performed per day (*Figure 1F*) and the proportion of individuals that were either presymptomatic, asymptomatic or recovered but continued to test positive for up to seven days, amongst all individuals on the ship not reporting symptoms. All testing was assumed to have 100% sensitivity and specificity.

Crew and passengers were modelled separately, using stratified data on the number of confirmed symptomatic cases (*Figure 1A–B*). We estimated the within-crew and within-passenger contact rates through calibration to the data, but assumed that the between-group contact rate was a fixed factor of 1/10th of the within-passenger rate, and explored the impact of this assumption in sensitivity analyses. We enabled the model to capture potential changes in contact behaviour between individuals by representing contact rates as sigmoid functions over time, reflecting any reductions in contact. The dates and extent of the changes were determined solely through model calibration to the data.

## Model parameterisation

We used data from the literature to inform the natural history of COVID-19, in particular for the duration of presymptomatic and symptomatic phases (see *Table 2*).

## Model calibration

The model was calibrated in a Bayesian framework. We fitted to the daily incidence of confirmed symptomatic cases with a known onset date, separately for passengers and crew, assuming a

**Table 2.** Model parameters and priors/values.

| Parameter | Description | Prior/value | Source/Notes |
|---|---|---|---|
| $\bar{\beta}$ | Overall contact rate (1/days) | Estimated: Uniform (0,100) | |
| $c^{(cc)}$ | Relative initial contact rate between crew/crew | Fixed: 1 | |
| $c^{(pp)}$ | Relative initial contact rate between passengers/passengers | Estimated: Uniform (0,100) | |
| $c^{(pc)}$ | Relative initial contact rate between passengers/crew | Fixed relative to $c^{(pp)}$ | |
| $X$ | Ratio: $\frac{c^{(pc)}}{c^{(pp)}}$ | Fixed: 0.1 | Assumed. Varied in sensitivity analyses |
| $b_1$ | Percentage reduction in all initial contact rates (%) | Estimated: Uniform (0,100) | |
| $b_2$ | Rate of change of all contact rates (1/days) | Fixed: 10 | Assumed. Transitions completed over approximately one day |
| $\tau^{(pp)}, \tau^{(pc)}$ | Time of transition for contacts between passengers/passengers and passengers/crew (days) | Estimated: Uniform(0,32) | Assumed to be equal to each other |
| $\tau^{(cc)}$ | Time of transition for contacts between crew/crew (days) | Estimated: Uniform(0,32) | |
| $\theta_p$ | Relative infectiousness of presymptomatic state | Estimated: Uniform(0,1) | Relative to symptomatic state |
| $\theta_a$ | Relative infectiousness of asymptomatic state | Estimated: Uniform(0,1) | Relative to symptomatic state |
| $\chi$ | Proportion of infections that proceed to asymptomatic state | Estimated: Uniform(0,1) | |
| $\frac{1}{\nu}$ | Latent period (days) | Fixed: 4.3 | Derived from *Backer et al., 2020* |
| $\frac{1}{\gamma_a}$ | Mean duration in asymptomatic state (days) | Fixed: 5.0 | Assumed. Sum of mean durations in presymptomatic and symptomatic states. Varied in sensitivity analyses. |
| $\frac{1}{\gamma_p}$ | Mean duration in presymptomatic state (days) | Fixed: 2.1 | Derived from *Backer et al., 2020* |
| $\frac{1}{\gamma_s}$ | Mean duration in infectious symptomatic state (days). | Fixed: 2.9 | From *Liu et al., 2020b* Applicable only until quarantine starts on 5th Feb |
| $\frac{1}{\mu}$ | Mean delay between onset of symptomatic disease and symptom-based testing and removal (days). | Fixed: 1 | Assumed. Applicable only after quarantine starts on 5th Feb. |
| $\frac{1}{\eta}$ | Mean duration of test positivity following recovery (days) | Fixed: 7 | From *Woelfel et al., 2020* |
| $\phi$ | Proportion of symptomatic cases with a reported onset date | Fixed: 0.661 (199/314) | From *Mizumoto et al., 2020*; *Nishiura, 2020* |
| $f(t)$ | Rate of symptom-agnostic testing and removal (1/days) | Fixed: Calculated | From *Mizumoto et al., 2020* Calculated using the number of tests administered per day amongst individuals not reporting symptoms (see Appendix 1) |
| $N^{(p)}$ | Total number of passengers on the ship as at start of quarantine on 5th Feb | Fixed: 2666 | From *NIID, 2020* |
| $N^{(c)}$ | Total number of crew on the ship as at start of quarantine on 5th Feb | Fixed: 1045 | From *NIID, 2020* |

Poisson distribution in the likelihood. We simultaneously fitted to the daily number of confirmed pre- and asymptomatic infections for passengers and crew combined by using the number of tests administered per day and the prevalence of presymptomatic, asymptomatic and post-infection test-positive individuals, assuming a binomial distribution in the likelihood. We used uniform priors for the parameters to be estimated (see *Table 2*) and sampled the posterior of the model parameters using sequential Markov Chain Monte-Carlo (MCMC). A burn in phase during which the proposal distributions were adapted in both scale and shape to provide optimal sampling efficiency was

discarded, leaving chains with 1.5 million iterations. The resultant MCMC chains were visually inspected for convergence.

## Model outputs

Model outputs were calculated by randomly sampling 100,000 parameter values from the posterior distribution. Model trajectories were generated and compared to the data in *Figure 1A–C* to inspect model fit. The basic reproduction number was also calculated over time using the next-generation matrix (*Diekmann et al., 2010*), as a measure of ongoing transmission. We estimated the proportion of infections that become asymptomatic and the relative infectiousness of asymptomatic infections using their respective marginal posterior parameter values. Finally, the contribution of asymptomatic infections to overall transmission, as well as the net reproduction number for presymptomatic passengers at the beginning of the outbreak (i.e. the typical number of infections generated by a single presymptomatic individual) were estimated, both overall and by specific ranges of relative infectiousness. We report the median and 95% equal-tailed posterior intervals throughout.

## Sensitivity analyses

We recalibrated the model for a number of alternative scenarios to assess model sensitivity. Firstly, we assessed the impact of removing the asymptomatic phase (i.e. 100% of infections progressed to symptomatic disease). Secondly, we explored the impact of assuming different values for the relative mixing between crew and passengers as well as shorter and longer durations of asymptomatic infection. Thirdly, we considered the impact of biased symptom-agnostic testing. Specifically, we first assumed that those that would test positive were 50% more likely to be tested, before then assuming that those that would test negative were 50% more likely to be tested, both compared to purely random testing as per the primary analysis. We also explored the impact of assuming a different proportion of asymptomatic infections for crew and passengers based on their distinct median ages (36 years for crew, 69 years for passengers), using a fixed ratio for the two proportions taken from the results of a model fitted to epidemic data in six countries by *Davies et al., 2020*. In addition, we explored a longer latent period given the relatively high age in our population (*Jiang et al., 2020*). Finally, we recalibrated the model assuming the 35 confirmed pre/asymptomatic cases where a test date was not available were allocated to the last feasible day (13th Feb) instead of proportionate to the overall number of tests over the period February 6-14th(see Appendix 2 for further details).

All analyses were conducted using R version 3.5.0 (*R Development Core Team, 2014*). Bayesian calibration was performed in LibBi (*Murray, 2013*) using RBi (*Funk, 2019*) as an interface. Replication data and analyses scripts are available on GitHub at https://github.com/thimotei/covid19_asymptomatic_trans (*Emery et al., 2020*; copy archived at https://github.com/elifesciences-publications/covid19_asymptomatic_trans).

## Role of funding source

The funder of the study had no role in study design, data collection, data analysis, data interpretation, or writing of the report. The corresponding author had full access to all the data in the study and had final responsibility for the decision to submit for publication.

## Acknowledgements

The authors would like to thank Dr Taichi Kidani for help in translating primary data sources.

The following funding sources are acknowledged as providing funding for the named authors. This research was partly funded by the Bill and Melinda Gates Foundation (INV-003174: YL; NTD Modelling Consortium OPP1184344: CABP). DFID/Wellcome Trust (Epidemic Preparedness Coronavirus research programme 221303/Z/20/Z: CABP). ERC Starting Grant (#757699: JCE, RMGJH). This project has received funding from the European Union's Horizon 2020 research and innovation programme - project EpiPose (101003688: YL). HDR UK (MR/S003975/1: RME). This research was partly funded by the National Institute for Health Research (NIHR) using UK aid from the UK Government to support global health research. The views expressed in this publication are those of the author(s) and not necessarily those of the NIHR or the UK Department of Health and Social Care (16/137/109: YL). UK MRC (MC_PC 19065: RME; MR/P014658/1: GMK). Wellcome Trust (206250/Z/17/Z: AJK, TWR; 208812/Z/17/Z: SFlasche; 210758/Z/18/Z: JH, SFunk).

The following funding sources are acknowledged as providing funding for the CMMID COVID-19 working group. Alan Turing Institute (AE). BBSRC LIDP (BB/M009513/1: DS). This research was partly funded by the Bill and Melinda Gates Foundation (INV-003174: KP, MJ; NTD Modelling Consortium OPP1184344: GM; OPP1180644: SRP; OPP1183986: ESN; OPP1191821: KO'R, MA). DFID/Wellcome Trust (Epidemic Preparedness Coronavirus research programme 221303/Z/20/Z: KvZ). Elrha R2HC/UK DFID/Wellcome Trust/This research was partly funded by the National Institute for Health Research (NIHR) using UK aid from the UK Government to support global health research. The views expressed in this publication are those of the author(s) and not necessarily those of the NIHR or the UK Department of Health and Social Care (KvZ). ERC Starting Grant (#757699: MQ). This project has received funding from the European Union's Horizon 2020 research and innovation programme - project EpiPose (101003688: KP, MJ, PK, WJE). This research was partly funded by the Global Challenges Research Fund (GCRF) project 'RECAP' managed through RCUK and ESRC (ES/P010873/1: AG, CIJ, TJ). Nakajima Foundation (AE). NIHR (16/137/109: BJQ, CD, FYS, MJ; Health Protection Research Unit for Modelling Methodology HPRU-2012–10096: NGD, TJ; PR-OD-1017–20002: AR). Royal Society (Dorothy Hodgkin Fellowship: RL; RP\EA\180004: PK). UK DHSC/UK Aid/NIHR (ITCRZ 03010: HPG). UK MRC (LID DTP MR/N013638/1: EMR, QJL). Authors of this research receive funding from UK Public Health Rapid Support Team funded by the United Kingdom Department of Health and Social Care (TJ). Wellcome Trust (208812/Z/17/Z: SC; 210758/Z/18/Z: JDM, NIB, SA, SRM). No funding (AKD, AMF, DCT, SH).

# Additional information

## Group author details

**CMMID COVID-19 Working Group**
Katherine E Atkins; Petra Klepac; Akira Endo; Christopher I Jarvis; Nicholas G Davies; Eleanor M Rees; Sophie R Meakin; Alicia Rosello; Kevin van Zandvoort; James D Munday; W John Edmunds; Thibaut Jombart; Megan Auzenbergs; Emily S Nightingale; Mark Jit; Sam Abbott; David Simons; Nikos I Bosse; Quentin J Leclerc; Simon R Procter; C Julian Villabona-Arenas; Damien C Tully; Arminder K Deol; Fiona Yueqian Sun; Stéphane Hué; Anna M Foss; Kiesha Prem; Graham Medley; Amy Gimma; Rachel Lowe; Samuel Clifford; Matthew Quaife; Charlie Diamond; Hamish P Gibbs; Billy J Quilty; Kathleen OReilly

## Funding

| Funder | Grant reference number | Author |
|---|---|---|
| European Research Council | Action Number 757699 | Jon C Emery<br>Rein MGJ Houben |
| Bill and Melinda Gates Foundation | INV-003174 | Yang Liu |
| Bill and Melinda Gates Foundation | NTD Modelling Consortium OPP1184344 | Carl AB Pearson |
| Department for International Development, UK Government | Epidemic Preparedness Coronavirus research programme 221303/Z/20/Z | Carl AB Pearson |
| Horizon 2020 | project EpiPose (101003688) | Yang Liu |
| HDR UK | MR/S003975/1 | Rosalind M Eggo |
| National Institute for Health Research | 16/137/109 | Yang Liu |
| Medical Research Council | MC_PC 19065 | Rosalind M Eggo |
| Medical Research Council | MR/P014658/1 | Gwenan M Knight |
| Wellcome | 206250/Z/17/Z | Timothy W Russell<br>Adam J Kucharski |
| Wellcome | 208812/Z/17/Z | Stefan Flasche |

| Wellcome | 210758/Z/18/Z | Joel Hellewell<br>Sebastian Funk |
|---|---|---|

The funders had no role in study design, data collection and interpretation, or the decision to submit the work for publication.

## Author contributions

Jon C Emery, Conceptualization, Data curation, Software, Formal analysis, Methodology, Writing - original draft, Writing - review and editing; Timothy W Russell, Joel Hellewell, Software, Formal analysis, Methodology, Writing - review and editing; Yang Liu, Data curation, Software, Methodology, Writing - review and editing; Carl AB Pearson, Sebastian Funk, Software, Methodology, Writing - review and editing; CMMID COVID-19 Working Group, Data curation, Writing - review and editing; Gwenan M Knight, Rosalind M Eggo, Adam J Kucharski, Methodology, Writing - review and editing; Stefan Flasche, Conceptualization, Methodology, Writing - review and editing; Rein MGJ Houben, Conceptualization, Methodology, Writing - original draft, Writing - review and editing

## Author ORCIDs

Jon C Emery (iD) https://orcid.org/0000-0001-6644-7604
Gwenan M Knight (iD) http://orcid.org/0000-0002-7263-9896
Adam J Kucharski (iD) http://orcid.org/0000-0001-8814-9421
Sebastian Funk (iD) http://orcid.org/0000-0002-2842-3406
Rein MGJ Houben (iD) https://orcid.org/0000-0003-4132-7467

## Decision letter and Author response

Decision letter https://doi.org/10.7554/eLife.58699.sa1
Author response https://doi.org/10.7554/eLife.58699.sa2

# Additional files

## Supplementary files

• Transparent reporting form

## Data availability

All data analysed during this study are included in the manuscript and supporting files. Model code and data are available through github at https://github.com/thimotei/covid19_asymptomatic_trans (copy archived at https://github.com/elifesciences-publications/covid19_asymptomatic_trans).

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

# Appendix 1

## Extended methods
### Data

Data for confirmed symptomatic cases were extracted from *Nishiura, 2020*. *Appendix 1—table 1* shows n = 199 confirmed symptomatic cases by date of symptom onset for passengers and crew separately. Symptom onset dates were unavailable for a further n = 115 confirmed symptomatic cases. These were accounted for in the model structure (see *Appendix 1—figure 1*) by assuming they were distributed over time proportional to those cases with a reported date of symptom onset. The data itself were not augmented.

**Appendix 1—table 1.** Confirmed symptomatic cases (n = 199) by date of symptom onset for passengers and crew separately, extracted from *Nishiura, 2020*.
A further n = 115 confirmed symptomatic cases without symptom onset dates are not included in the table.

| Date of symptom onset | Confirmed symptomatic cases | | |
| --- | --- | --- | --- |
| | Passengers | Crew | Total |
| 20-Jan | 2 | 0 | 2 |
| 21-Jan | 0 | 0 | 0 |
| 22-Jan | 0 | 0 | 0 |
| 23-Jan | 1 | 0 | 1 |
| 24-Jan | 0 | 0 | 0 |
| 25-Jan | 0 | 0 | 0 |
| 26-Jan | 0 | 0 | 0 |
| 27-Jan | 0 | 0 | 0 |
| 28-Jan | 0 | 0 | 0 |
| 29-Jan | 1 | 0 | 1 |
| 30-Jan | 1 | 0 | 1 |
| 31-Jan | 0 | 0 | 0 |
| 01-Feb | 4 | 0 | 4 |
| 02-Feb | 4 | 0 | 4 |
| 03-Feb | 4 | 0 | 4 |
| 04-Feb | 6 | 0 | 6 |
| 05-Feb | 12 | 0 | 12 |
| 06-Feb | 15 | 2 | 17 |
| 07-Feb | 29 | 2 | 31 |
| 08-Feb | 17 | 2 | 19 |
| 09-Feb | 19 | 5 | 24 |
| 10-Feb | 7 | 3 | 10 |
| 11-Feb | 11 | 8 | 19 |
| 12-Feb | 5 | 7 | 12 |
| 13-Feb | 9 | 8 | 17 |
| 14-Feb | 2 | 5 | 7 |
| 15-Feb | 1 | 3 | 4 |
| 16-Feb | 0 | 3 | 3 |
| 17-Feb | 0 | 1 | 1 |

*Continued on next page*

*Appendix 1—table 1 continued*

| Date of symptom onset | Confirmed symptomatic cases | | |
|---|---|---|---|
| | Passengers | Crew | Total |
| 18-Feb | 0 | 0 | 0 |
| 19-Feb | 0 | 0 | 0 |
| 20-Feb | 0 | 0 | 0 |
| Total | 150 | 49 | 199 |

Data for confirmed pre/asymptomatic cases and symptom-agnostic testing were extracted from *Mizumoto et al., 2020*. *Appendix 1—table 2* shows n = 2749 symptom-agnostic tests and n = 320 confirmed pre/asymptomatic cases by date of test for passengers and crew combined, since stratification by passenger/crew was unavailable. The number of symptom-agnostic tests was inferred from the total number of tests each day, minus the number of positive results in individuals reporting symptoms in *Mizumoto et al., 2020*. Test dates were not available for n = 35 confirmed pre/asymptomatic cases between 5th-14th Feb. These were distributed proportional to the total number of tests (symptom-based and symptom-agnostic) on those days. An alternative scenario where all n = 35 confirmed pre/asymptomatic cases are allocated to the last possible day (13th Feb) is explored in sensitivity analyses.

**Appendix 1—table 2.** Confirmed pre/asymptomatic cases (n = 320) and symptom-agnostic tests (n = 2749) by date of test for passengers and crew combined, extracted from *Mizumoto et al., 2020*.
[+]Test dates were not available for n = 35 confirmed pre/asymptomatic cases between 5th-14th Feb. These were distributed proportional to the total number of tests (symptom-based and symptom-agnostic) on those days.

| Date of test | Number of symptom agnostic tests | Number of confirmed pre/asymptomatic cases |
|---|---|---|
| 20-Jan | 0 | 0 |
| 21-Jan | 0 | 0 |
| 22-Jan | 0 | 0 |
| 23-Jan | 0 | 0 |
| 24-Jan | 0 | 0 |
| 25-Jan | 0 | 0 |
| 26-Jan | 0 | 0 |
| 27-Jan | 0 | 0 |
| 28-Jan | 0 | 0 |
| 29-Jan | 0 | 0 |
| 30-Jan | 0 | 0 |
| 31-Jan | 0 | 0 |
| 01-Feb | 0 | 0 |
| 02-Feb | 0 | 0 |
| 03-Feb | 0 | 0 |
| 04-Feb | 0 | 0 |
| 05-Feb[+] | 23 | 2 |
| 06-Feb[+] | 64 | 3 |
| 07-Feb[+] | 138 | 8 |
| 08-Feb[+] | 3 | 0 |
| 09-Feb[+] | 54 | 3 |

*Continued on next page*

*Appendix 1—table 2 continued*

| Date of test | Number of symptom agnostic tests | Number of confirmed pre/asymptomatic cases |
|---|---|---|
| 10-Feb[+] | 43 | 5 |
| 11-Feb[+] | 0 | 0 |
| 12-Feb[+] | 17 | 3 |
| 13-Feb[+] | 188 | 11 |
| 14-Feb[+] | 0 | 0 |
| 15-Feb | 188 | 38 |
| 16-Feb | 257 | 38 |
| 17-Feb | 475 | 70 |
| 18-Feb | 658 | 65 |
| 19-Feb | 596 | 68 |
| 20-Feb | 45 | 6 |
| Total | 2749 | 320 |

## Model

The model described in the main article is shown in detail in *Appendix 1—figure 1*, where passengers ($i = p$) and crew ($i = c$) are modelled separately and the annotated parameters are described in *Table 2* in the main article.

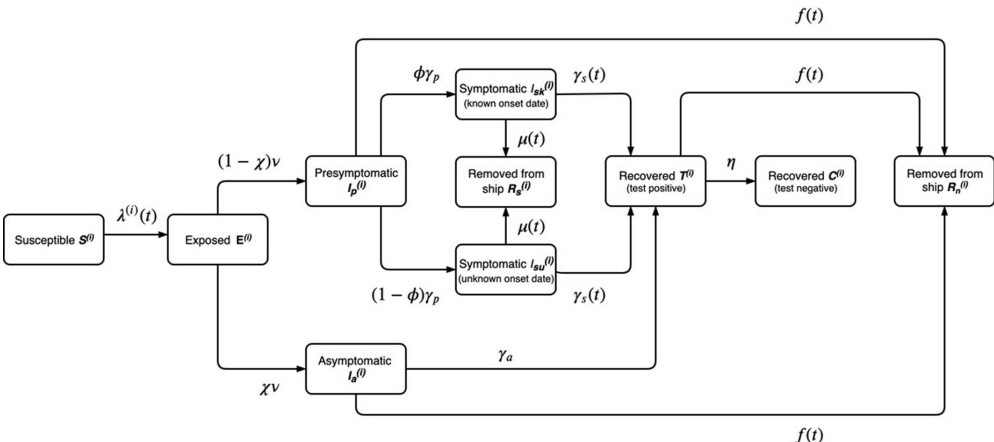

**Appendix 1—figure 1.** Model diagram for the outbreak onboard the Diamond Princess cruise ship described in the main paper. The annotated transition parameters are defined in *Table 2* in the main article and detailed further, below. The model is stratified by i = passengers or crew. The asymptomatic, presymptomatic and symptomatic states are all assumed to be infectious and individuals would test positive during symptom-based or symptom-agnostic testing. Individuals that recover are also assumed to test positive for an average of 1 week after they are no longer infectious.

The force of infection is given by

$$\lambda^{(i)}(t) = \sum_{j=p,c} \beta^{(ij)}(t) \frac{\left(\theta_a I_a^{(j)} + \theta_p I_p^{(j)} + I_{sk}^{(j)} + I_{su}^{(j)}\right)}{N^{(j)}}$$

where the time dependent contact parameters are described by sigmoid functions

$$\beta^{(ij)}(t) = \bar{\beta} c^{(ij)} \left(1 - \frac{b_1}{1 + e^{-b_2(t - \tau^{(ij)})}}\right)$$

And $\tau^{(pp)} = \tau^{(pc)} = \tau^{(cp)}$ (i.e. contact between passengers/passengers and passengers/crew is reduced at the same time, which can differ from contact between crew/crew).

The transition from exposed to presymptomatic or asymptomatic is modelled as an erlang distribution using two compartments (i.e. a shape parameter $k = 2$), each with a mean duration of $1/2v$.

The rate of symptom agnostic testing and removal of individuals not reporting symptoms is given by the total number of symptom agnostic tests administered per day divided by the total number of individuals not presenting symptoms being tested on that day

$$f(t) = \frac{N^{\text{tests}}}{(S + E + I_a + I_p + T + C)}$$

Where $N^{\text{tests}}$ is taken from the data in **Appendix 1—table 2** and variables without indices represent the totals among passengers and crew (e.g. $S = S^{(p)} + S^{(c)}$)

To reflect heightened symptom awareness following quarantine, the transition rate from symptomatic infection to recovered on the ship is constant before quarantine and zero afterwards, whilst the rate of removal of individuals reporting symptoms is zero before quarantine and a constant afterwards

$$\gamma_s(t) = \gamma_s, \; \mu(t) = 0 \quad \text{for } t < 5\text{th Feb}$$
$$\gamma_s(t) = 0, \; \mu(t) = \mu \quad \text{for } t \geq 5\text{th Feb}$$

All other model transitions are exponentially distributed.

The model is initialised with a single symptomatic passenger with a known onset date on 20th Jan, with all other individuals susceptible

$$I_{sk}^{(p)}(0) = 1, \quad S^{(p)} = N^{(p)} - 1, \quad S^{(c)} = N^{(c)}$$

## Model calibration

The model was calibrated in a Bayesian framework to fit to the two sets of empirical observations from the ship (**Appendix 1—tables 1–2**). We used a Poisson likelihood for the incident symptomatic cases with a known onset date for crew and passengers separately. We used a Binomial likelihood for the number of confirmed pre- and asymptomatic infections for passengers and crew combined, using the number of tests administered per day and the prevalence of presymptomatic, asymptomatic and post-infection test-positive individuals. The complete likelihood is given by

$$L = \left( \prod_{k=1}^{K} \text{Poisson}\left(Z_k^{(c)} | \text{mean} = z_k^{(c)}\right) \right) \left( \prod_{k=1}^{K} \text{Poisson}\left(Z_k^{(p)} | \text{mean} = z_k^{(p)}\right) \right) \left( \prod_{k=1}^{K} \text{Binom}\left(Y_k | N_k^{\text{tests}}, \text{mean} = y_k\right) \right)$$

where $Z_k^{(i)}$ is the observed incidence of symptomatic cases with a known date of onset on day $k$ for passengers $p$ or crew $c$, $z_k^{(i)}$ is the model predicted incidence, $Y_k$ is the observed prevalence of presymptomatic, asymptomatic and post-infection test-positive individuals (passengers and crew combined) amongst $N_k^{\text{tests}}$ symptom-agnostic tests, and $y_k$ is the model predicted prevalence

$$y(t) = \frac{I_a + I_p + T}{S + E + I_a + I_p + T + C}$$

We used uniform priors for the parameters to be estimated (see **Table 2** in the main article).

## Model outputs

The basic reproduction number as a function of time $R_0(t)$ was calculated by first constructing the next generation matrix (NGM) at each time point using the relevant Jacobian matrices (**Diekmann et al., 2010**). The basic reproduction number is then given by the absolute value of the dominant eigenvalue of the NGM.

The net reproduction number for a presymptomatic infection (i.e. the typical number of secondary infections caused by a single presymptomatic individual throughout both their presymptomatic

and symptomatic periods) at the beginning of the outbreak is given by the respective entry in the NGM evaluated at $t = 0$.

The instantaneous proportion of transmission from either symptomatic, presymptomatic or asymptomatic individuals was calculated by dividing the number of infections generated by the respective infected state in the previous timestep by the total number of new infections in the previous timestep. The overall proportion of transmission from asymptomatic individuals was given by the cumulative number of infections caused by asymptomatics divided by the cumulative number of total infections, evaluated at the end of the outbreak.

## Appendix 2

### Sensitivity analyses results

1. Presymptomatic infection only

Assumes the proportion of infections that are asymptomatic and their relative infectiousness are zero ($\chi = 0$ and $\theta_a = 0$). The latent period $1/v$ is estimated with a uniform prior between 1 and 21 days.

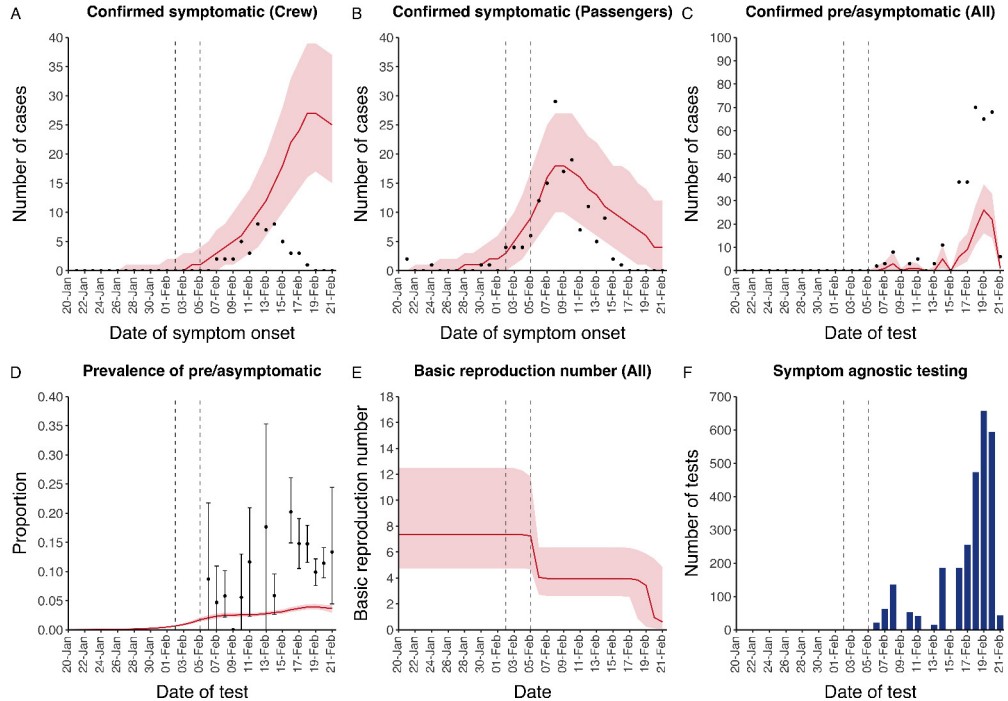

**Appendix 2—figure 1.** Data from the Diamond Princess and model calibration. Figure shows data from the Diamond Princess (points (**A–D**) and bars (**F**)) and results from model calibration. Red lines = median, shading = 95% posterior plus observational interval (**A–C**) and 95% posterior interval only (**D–E**). Two vertical lines show the date of the first confirmed diagnosis (left) and the start of quarantine measures (right). (**A–B**) show confirmed symptomatic cases among crew (**A**) and passengers (**B**) with a reported date of onset; (**C**) shows confirmed pre- or asymptomatic individuals by test date; (**D**) shows the prevalence of pre/asymptomatic individuals by test date. Points and error bars show point estimates and 95% confidence intervals; (**E**) shows the basic reproduction number over time for the ship as a whole, reflecting the drop in contact rates (**F**) shows the number of tests administered irrespective of symptoms, by test date.

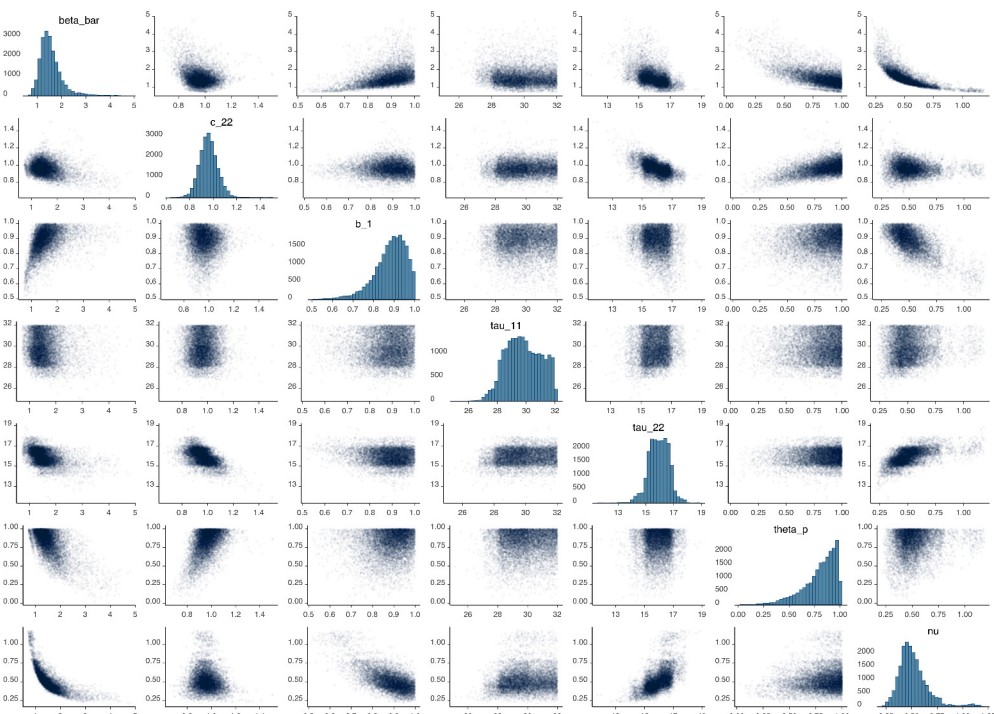

**Appendix 2—figure 2.** Parameter correlation plot containing parameter values from 10,000 samples of the joint posterior distribution found during MCMC model calibration. See *Table 2* in the main article for parameter definitions and descriptions.

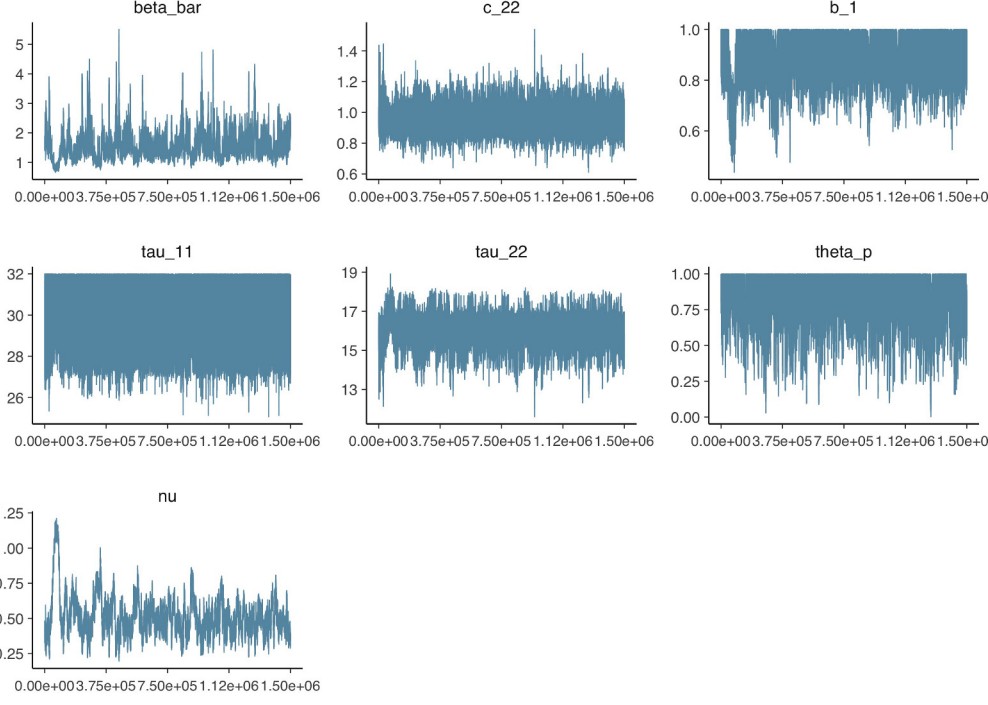

**Appendix 2—figure 3.** Parameter trace plot showing all 1.5 million samples from the MCMC model calibration sequentially. See *Table 2* in the main article for parameter definitions and descriptions.

## 2. Relative passenger-crew contact rate: X = 0.02

Assumes the contact rate between passengers and crew is 1/50th of contacts between passengers and passengers $\left(c^{(pc)}/c^{(pp)} = 0.02\right)$.

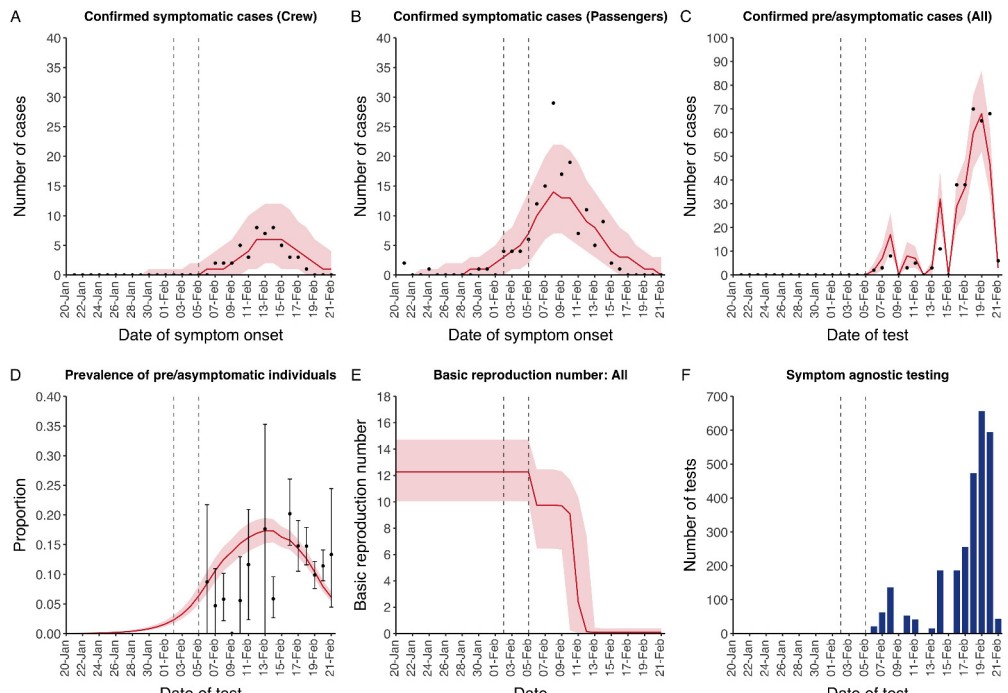

**Appendix 2—figure 4.** Data from the Diamond Princess and model calibration. Figure shows data from the Diamond Princess (points (**A–D**) and bars (**F**)) and results from model calibration. Red lines = median, shading = 95% posterior plus observational interval (**A–C**) and 95% posterior interval only (**D–E**). Two vertical lines show the date of the first confirmed diagnosis (left) and the start of quarantine measures (right). (**A–B**) show confirmed symptomatic cases among crew (**A**) and passengers (**B**) with a reported date of onset; (**C**) shows confirmed pre- or asymptomatic individuals by test date; (**D**) shows the prevalence of pre/asymptomatic individuals by test date. Points and error bars show point estimates and 95% confidence intervals; (**E**) shows the basic reproduction number over time for the ship as a whole, reflecting the drop in contact rates (**F**) shows the number of tests administered irrespective of symptoms, by test date.

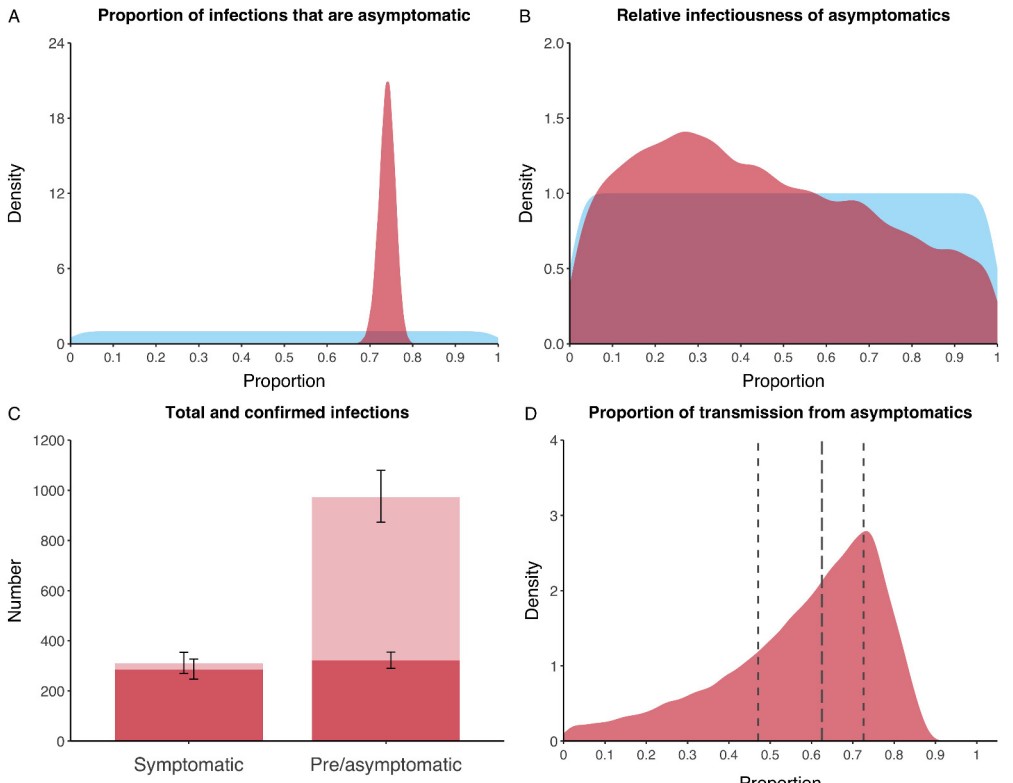

**Appendix 2—figure 5.** Proportion of infections that are asymptomatic and their contribution to transmission. (**A**) Prior (blue) and posterior (red) probability distribution for the proportion progressing to asymptomatic infections. (**B**) Prior (blue) and posterior (red) probability distribution for the relative infectiousness of asymptomatic infections. (**C**) Number of pre- and asymptomatic infections and symptomatic cases detected (dark red) and not detected (light red) during the outbreak. Error bars indicate 95% posterior intervals. (**D**) Posterior probability distribution for proportion of transmission that is from asymptomatic individuals. Dashed and dotted lines show median and interquartile range respectively.

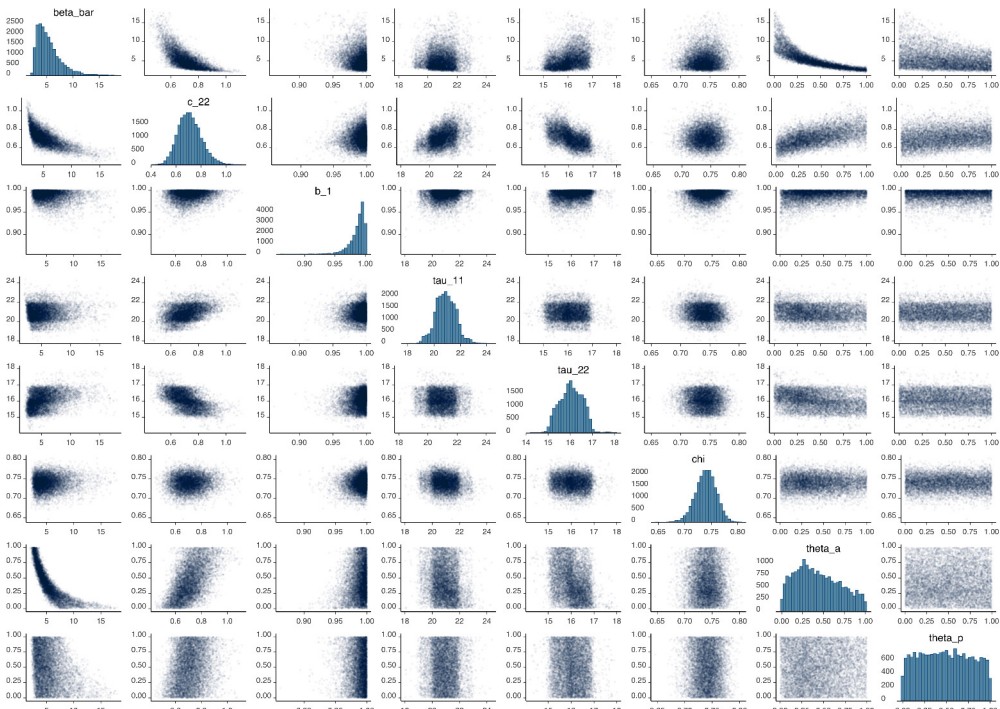

**Appendix 2—figure 6.** Parameter correlation plot containing parameter values from 10,000 samples of the joint posterior distribution found during MCMC model calibration. See *Table 2* in the main article for parameter definitions and descriptions.

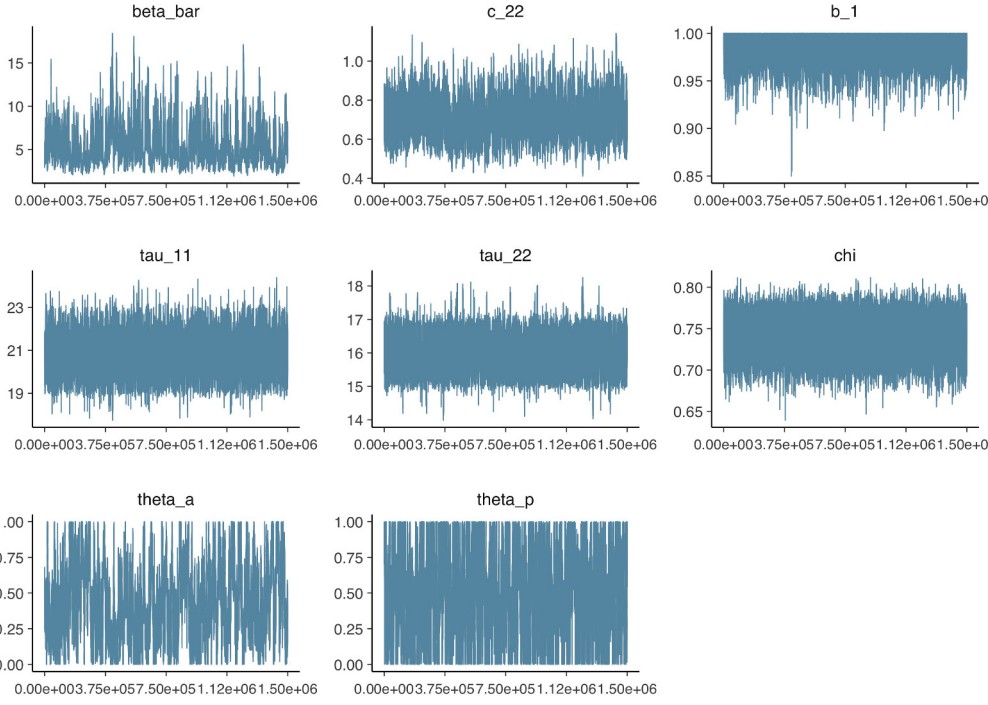

**Appendix 2—figure 7.** Parameter trace plot showing all 1.5 million samples from the MCMC model calibration sequentially. See *Table 2* in the main article for parameter definitions and descriptions.

## 3. Relative passenger-crew contact rate: X = 0.5

Assumes the contact rates between passengers and crew is half that of between passengers and passengers $(c^{(pc)}/c^{(pp)} = 0.5)$.

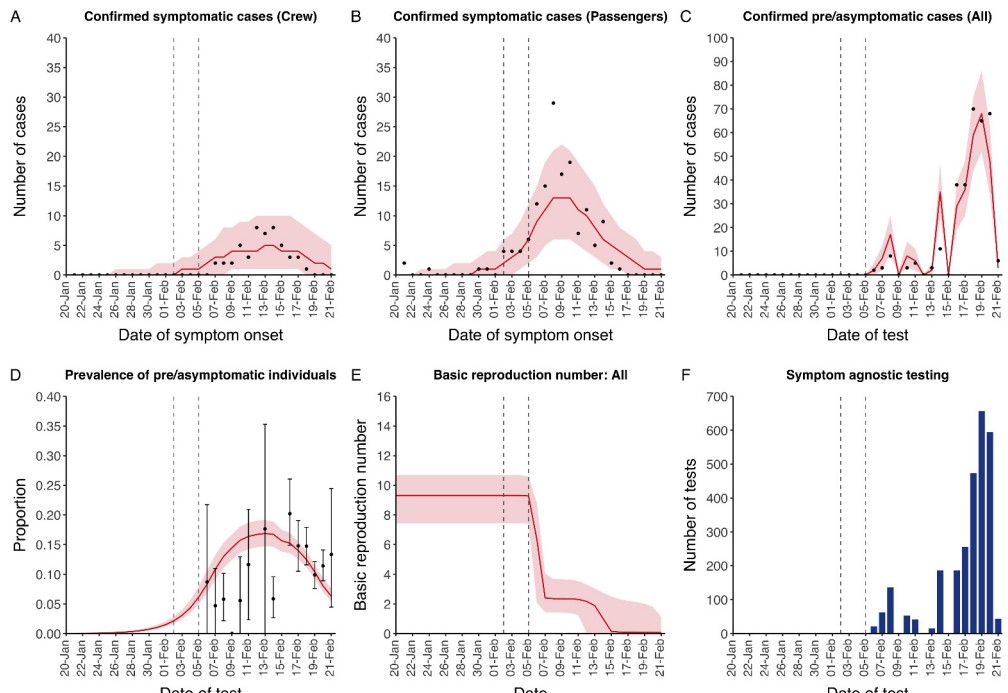

**Appendix 2—figure 8.** Data from the Diamond Princess and model calibration. Figure shows data from the Diamond Princess (points (**A–D**) and bars (**F**)) and results from model calibration. Red lines = median, shading = 95% posterior plus observational interval (**A–C**) and 95% posterior interval only (**D–E**). Two vertical lines show the date of the first confirmed diagnosis (left) and the start of quarantine measures (right). (**A–B**) show confirmed symptomatic cases among crew (**A**) and passengers (**B**) with a reported date of onset; (**C**) shows confirmed pre- or asymptomatic individuals by test date; (**D**) shows the prevalence of pre/asymptomatic individuals by test date. Points and error bars show point estimates and 95% confidence intervals; (**E**) shows the basic reproduction number over time for the ship as a whole, reflecting the drop in contact rates (**F**) shows the number of tests administered irrespective of symptoms, by test date.

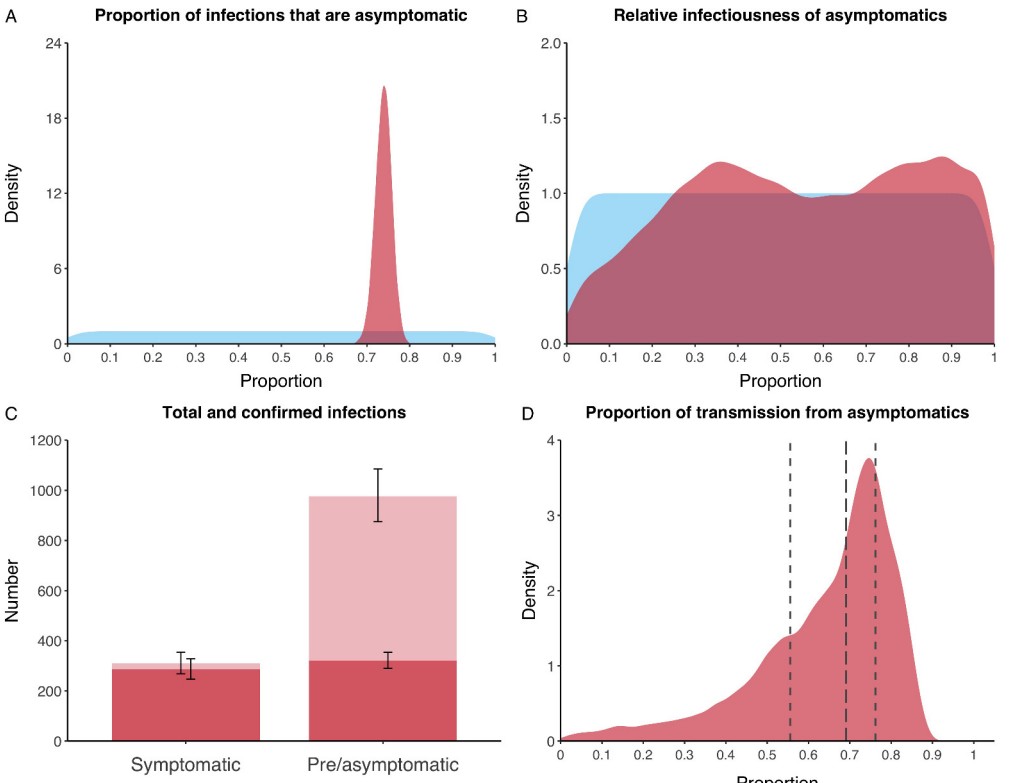

**Appendix 2—figure 9.** Proportion of infections that are asymptomatic and their contribution to transmission. (**A**) Prior (blue) and posterior (red) probability distribution for the proportion progressing to asymptomatic infections. (**B**) Prior (blue) and posterior (red) probability distribution for the relative infectiousness of asymptomatic infections. (**C**) Number of pre- and asymptomatic infections and symptomatic cases detected (dark red) and not detected (light red) during the outbreak. Error bars indicate 95% posterior intervals. (**D**) Posterior probability distribution for proportion of transmission that is from asymptomatic individuals. Dashed and dotted lines show median and interquartile range respectively.

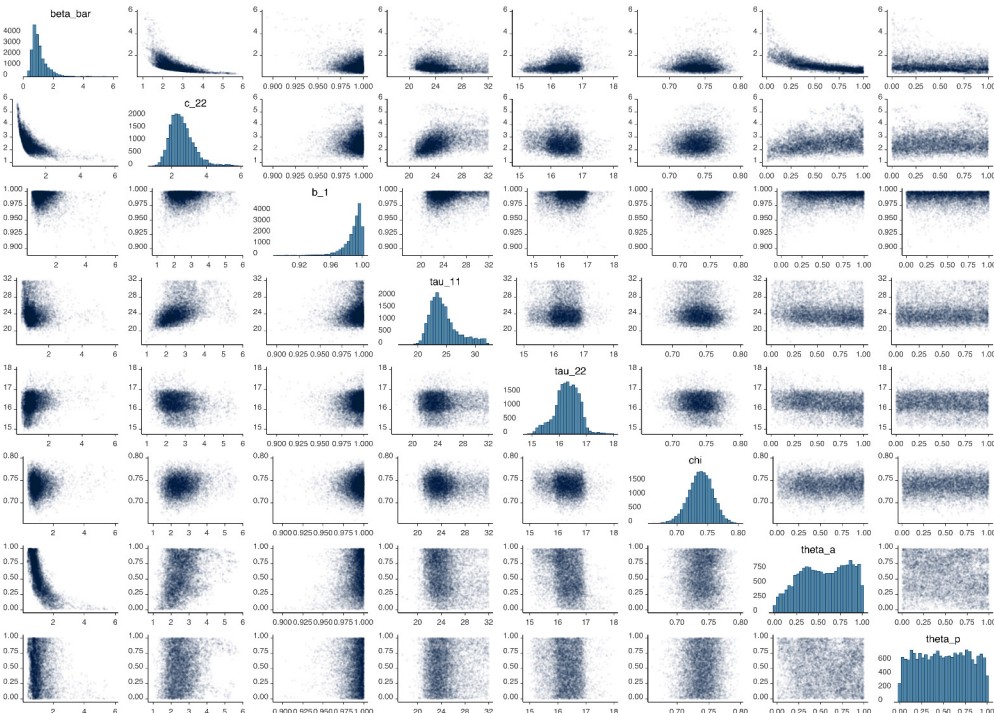

**Appendix 2—figure 10.** Parameter correlation plot containing parameter values from 10,000 samples of the joint posterior distribution found during MCMC model calibration. See *Table 2* in the main article for parameter definitions and descriptions.

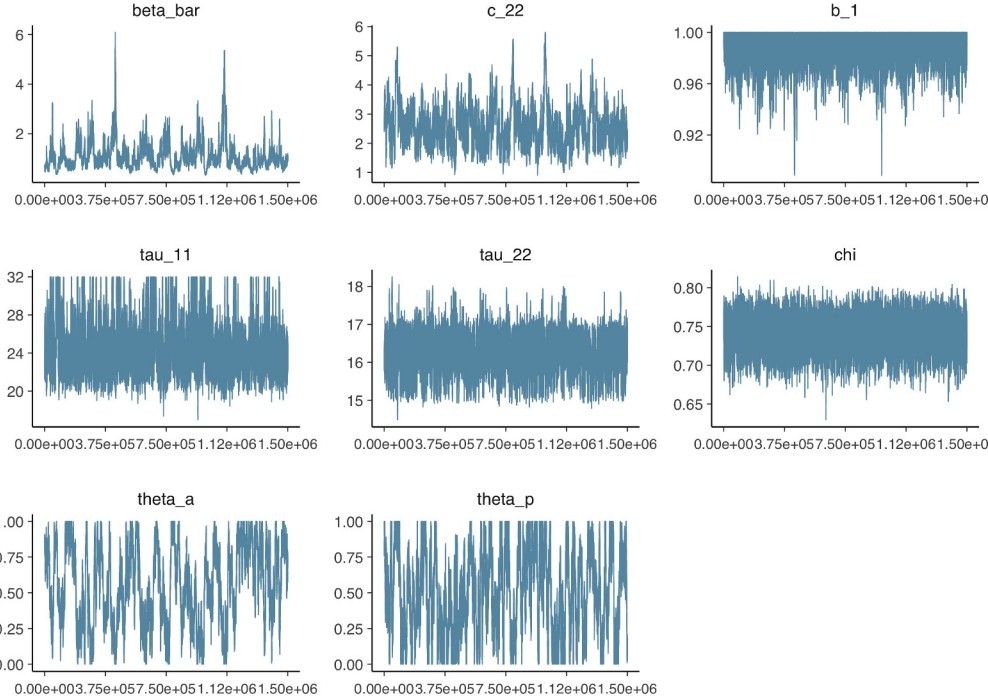

**Appendix 2—figure 11.** Parameter trace plot showing all 1.5 million samples from the MCMC model calibration sequentially. See *Table 2* in the main article for parameter definitions and descriptions.

## 4. Biased symptom-agnostic testing: test-negative individuals more likely to be tested

Assumes individuals that would test negative during symptom-agnostic testing are 50% more likely to be tested compared to the primary analysis, where testing is random.

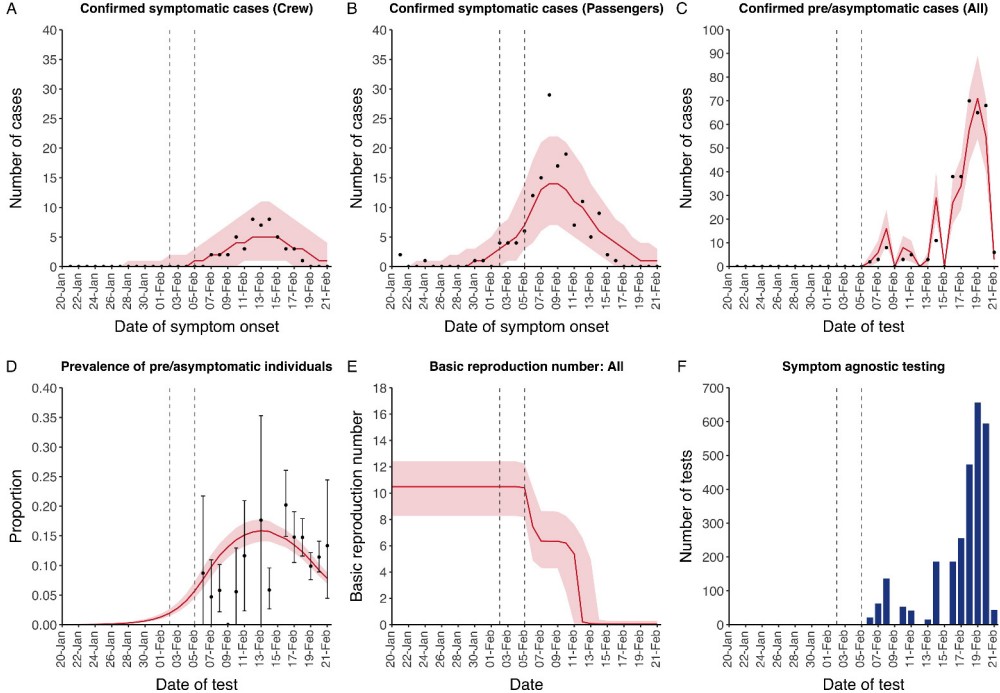

**Appendix 2—figure 12.** Data from the Diamond Princess and model calibration. Figure shows data from the Diamond Princess (points (**A–D**) and bars (**F**)) and results from model calibration. Red lines = median, shading = 95% posterior plus observational interval (**A–C**) and 95% posterior interval only (**D–E**). Two vertical lines show the date of the first confirmed diagnosis (left) and the start of quarantine measures (right). (**A–B**) show confirmed symptomatic cases among crew (**A**) and passengers (**B**) with a reported date of onset; (**C**) shows confirmed pre- or asymptomatic individuals by test date; (**D**) shows the prevalence of pre/asymptomatic individuals by test date. Points and error bars show point estimates and 95% confidence intervals; (**E**) shows the basic reproduction number over time for the ship as a whole, reflecting the drop in contact rates (**F**) shows the number of tests administered irrespective of symptoms, by test date.

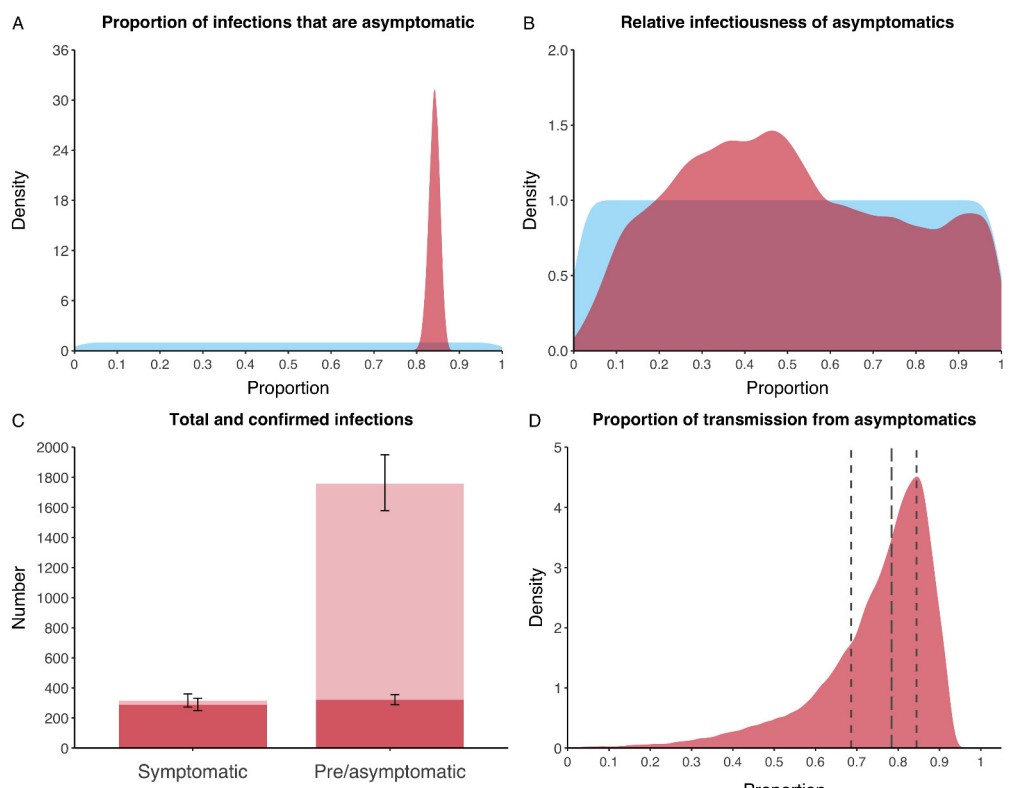

**Appendix 2—figure 13.** Proportion of infections that are asymptomatic and their contribution to transmission. (**A**) Prior (blue) and posterior (red) probability distribution for the proportion progressing to asymptomatic infections. (**B**) Prior (blue) and posterior (red) probability distribution for the relative infectiousness of asymptomatic infections. (**C**) Number of pre- and asymptomatic infections and symptomatic cases detected (dark red) and not detected (light red) during the outbreak. Error bars indicate 95% posterior intervals. (**D**) Posterior probability distribution for proportion of transmission that is from asymptomatic individuals. Dashed and dotted lines show median and interquartile range respectively.

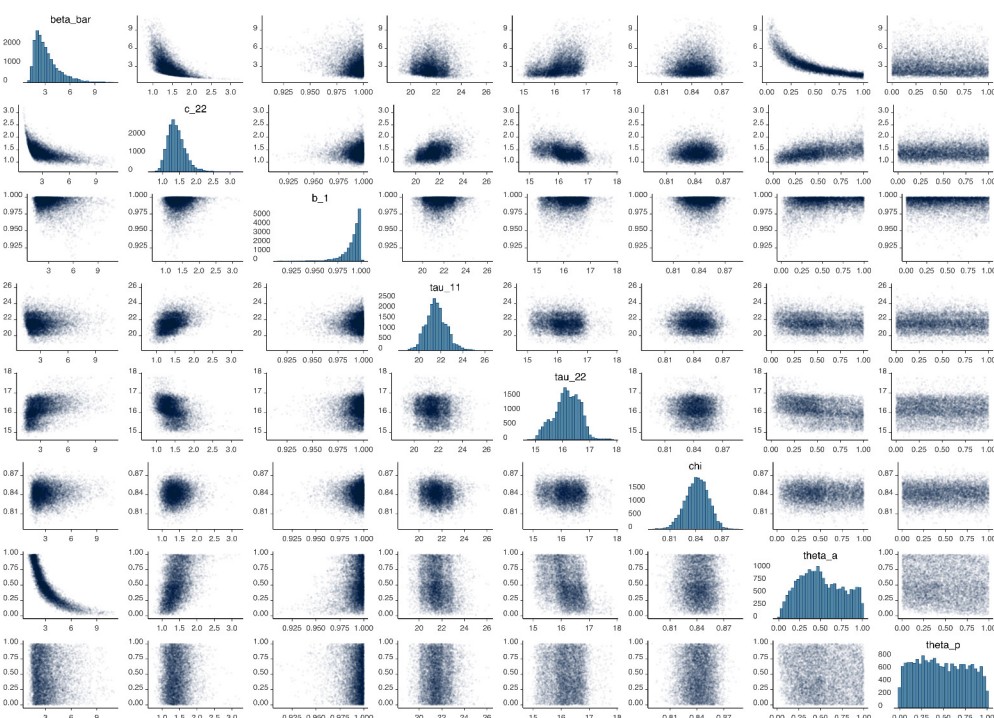

**Appendix 2—figure 14.** Parameter correlation plot containing parameter values from 10,000 samples of the joint posterior distribution found during MCMC model calibration. See *Table 2* in the main article for parameter definitions and descriptions.

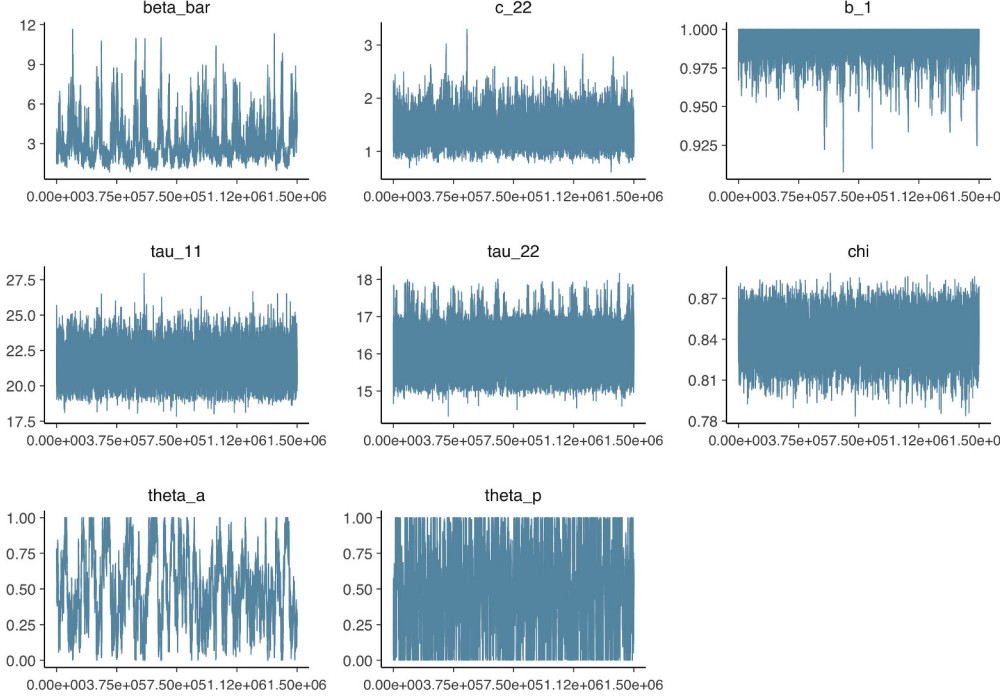

**Appendix 2—figure 15.** Parameter trace plot showing all 1.5 million samples from the MCMC model calibration sequentially. See *Table 2* in the main article for parameter definitions and descriptions.

## 5. Biased symptom-agnostic testing: test-positive individuals more likely to be tested

Assumes individuals that would test positive during symptom-agnostic testing are 50% more likely to be tested compared to the primary analysis, where testing is random.

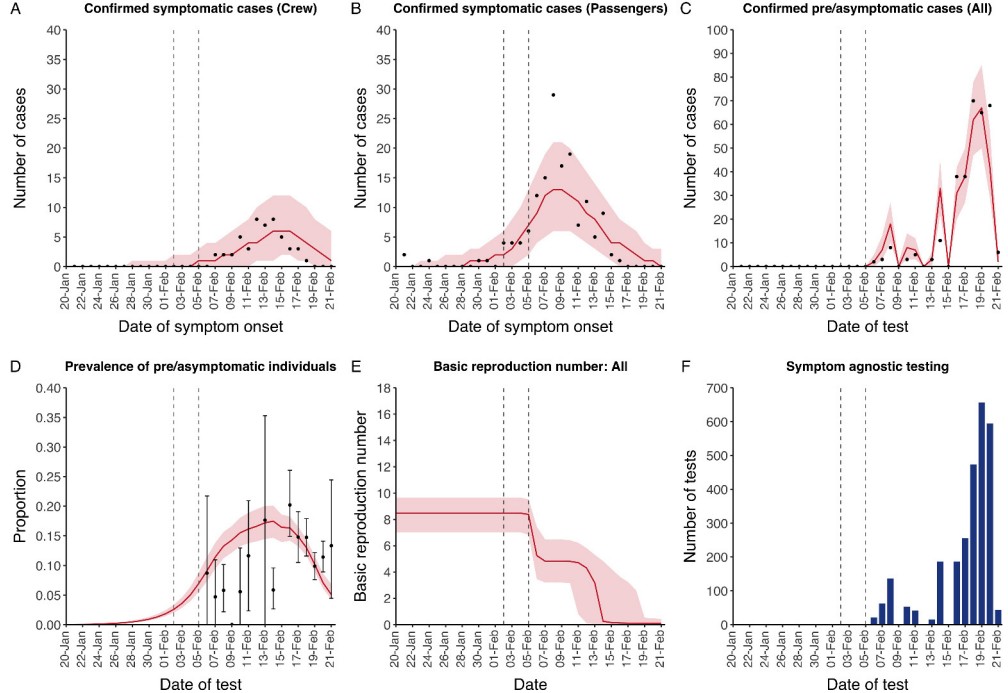

**Appendix 2—figure 16.** Data from the Diamond Princess and model calibration. Figure shows data from the Diamond Princess (points (**A–D**) and bars (**F**)) and results from model calibration. Red lines = median, shading = 95% posterior plus observational interval (**A–C**) and 95% posterior interval only (**D–E**). Two vertical lines show the date of the first confirmed diagnosis (left) and the start of quarantine measures (right). (**A–B**) show confirmed symptomatic cases among crew (**A**) and passengers (**B**) with a reported date of onset; (**C**) shows confirmed pre- or asymptomatic individuals by test date; (**D**) shows the prevalence of pre/asymptomatic individuals by test date. Points and error bars show point estimates and 95% confidence intervals; (**E**) shows the basic reproduction number over time for the ship as a whole, reflecting the drop in contact rates (**F**) shows the number of tests administered irrespective of symptoms, by test date.

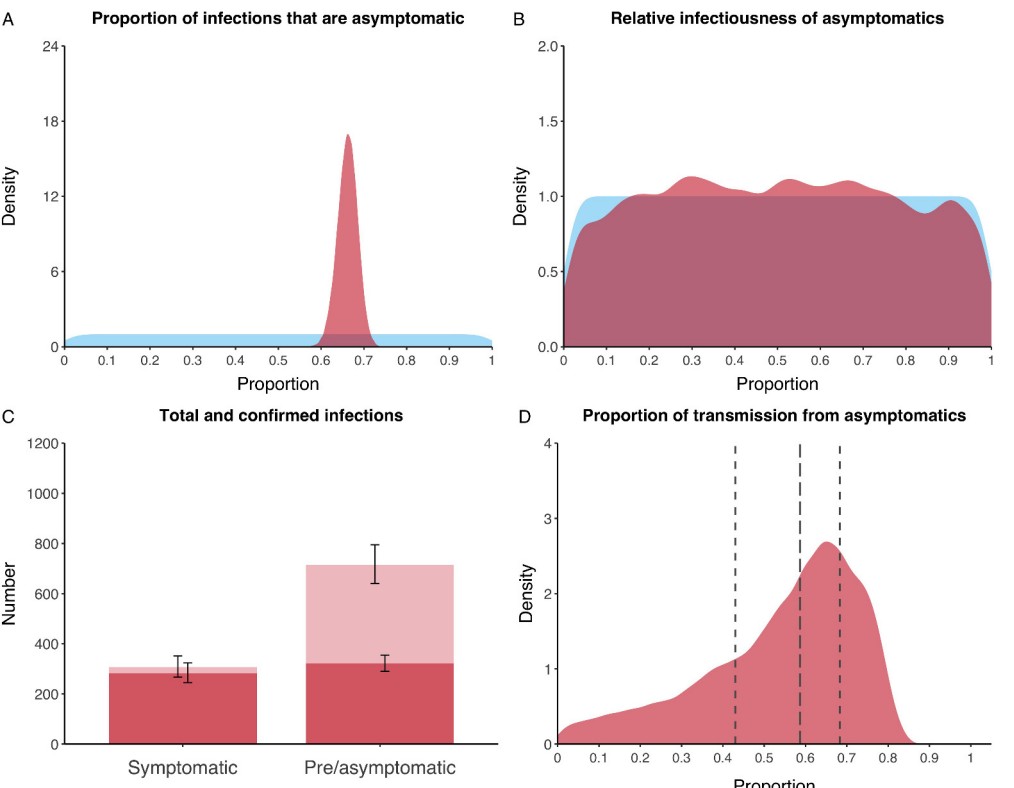

**Appendix 2—figure 17.** Proportion of infections that are asymptomatic and their contribution to transmission. (**A**) Prior (blue) and posterior (red) probability distribution for the proportion progressing to asymptomatic infections. (**B**) Prior (blue) and posterior (red) probability distribution for the relative infectiousness of asymptomatic infections. (**C**) Number of pre- and asymptomatic infections and symptomatic cases detected (dark red) and not detected (light red) during the outbreak. Error bars indicate 95% posterior intervals. (**D**) Posterior probability distribution for proportion of transmission that is from asymptomatic individuals. Dashed and dotted lines show median and interquartile range respectively.

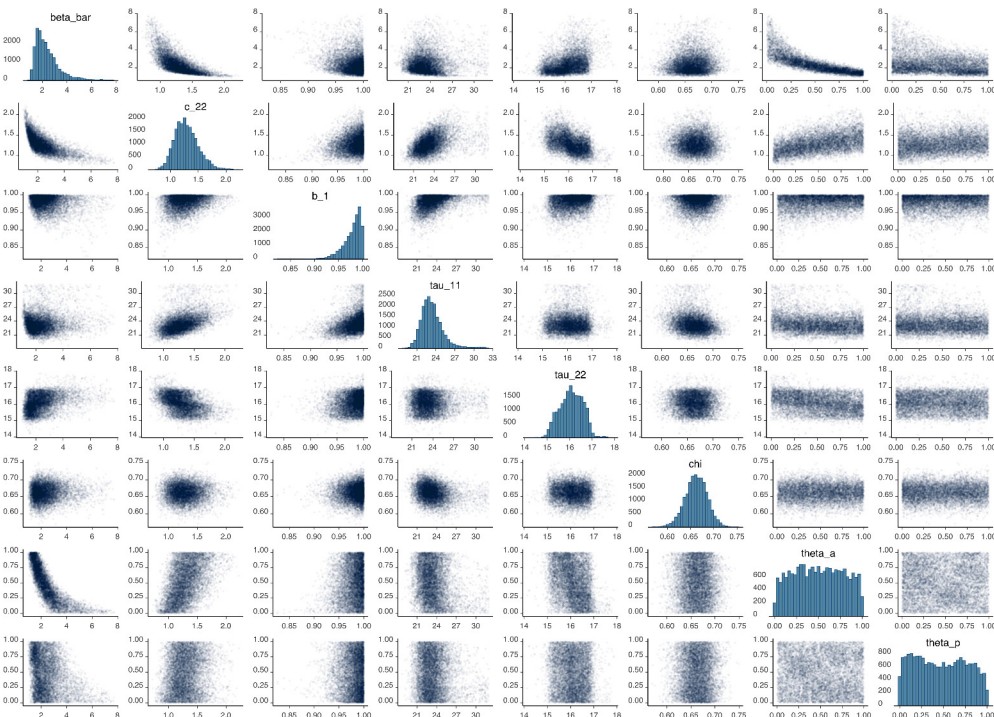

**Appendix 2—figure 18.** Parameter correlation plot containing parameter values from 10,000 samples of the joint posterior distribution found during MCMC model calibration. See *Table 2* in the main article for parameter definitions and descriptions.

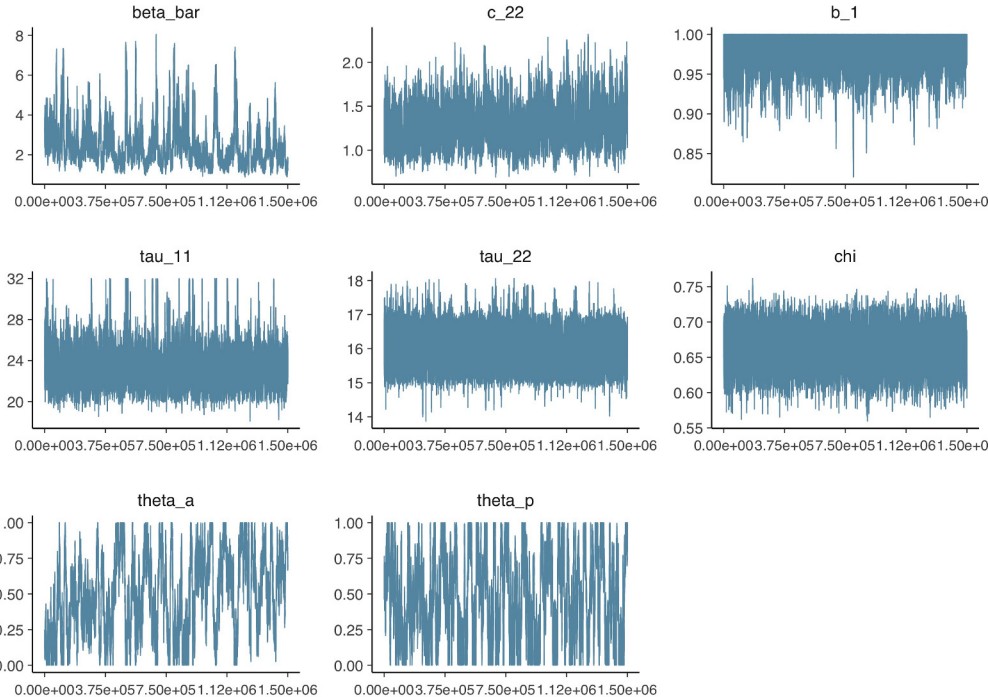

**Appendix 2—figure 19.** Parameter trace plot showing all 1.5 million samples from the MCMC model calibration sequentially. See *Table 2* in the main article for parameter definitions and descriptions.

## 6. Age dependent proportion asymptomatic

Assumes separate asymptomatic proportions for crew $(\chi^{(c)})$ and passengers $(\chi^{(p)})$ to reflect their different age demographics (median ages of 36 and 69 respectively), compared to a single asymptomatic proportion in the primary analysis. The ratio $\chi^{(p)}/\chi^{(c)}$ was fixed at 0.48 using the results for asymptomatic proportion by age from a model fitted to epidemic data in six countries by *Davies et al., 2020*.

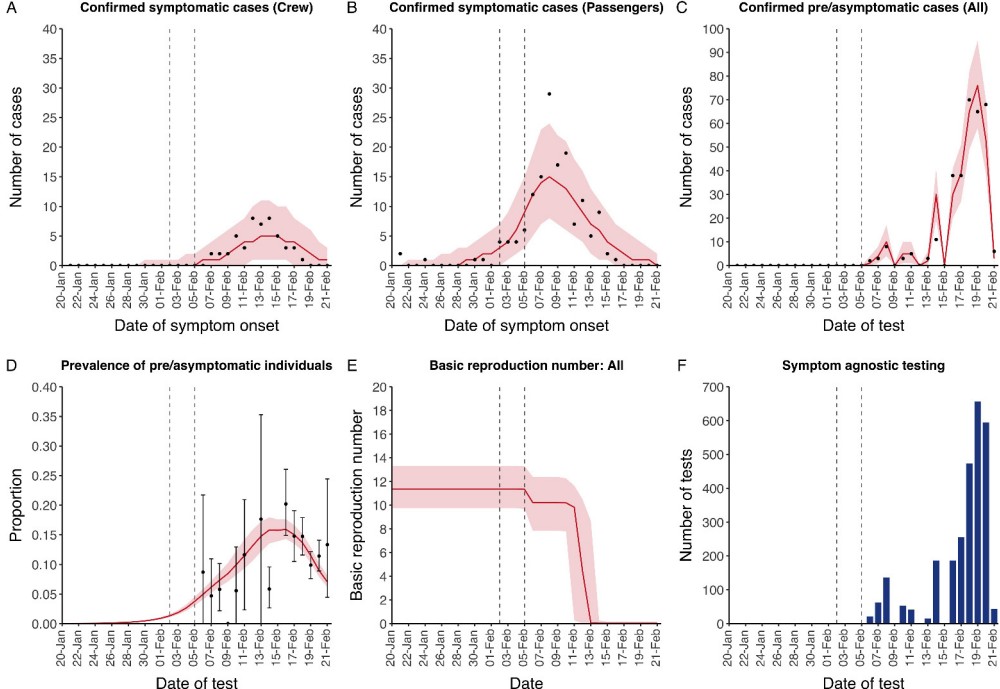

**Appendix 2—figure 20.** Data from the Diamond Princess and model calibration. Figure shows data from the Diamond Princess (points (**A–D**) and bars (**F**)) and results from model calibration. Red lines = median, shading = 95% posterior plus observational interval (**A–C**) and 95% posterior interval only (**D–E**). Two vertical lines show the date of the first confirmed diagnosis (left) and the start of quarantine measures (right). (**A–B**) show confirmed symptomatic cases among crew (**A**) and passengers (**B**) with a reported date of onset; (**C**) shows confirmed pre- or asymptomatic individuals by test date; (**D**) shows the prevalence of pre/asymptomatic individuals by test date. Points and error bars show point estimates and 95% confidence intervals; (**E**) shows the basic reproduction number over time for the ship as a whole, reflecting the drop in contact rates (**F**) shows the number of tests administered irrespective of symptoms, by test date.

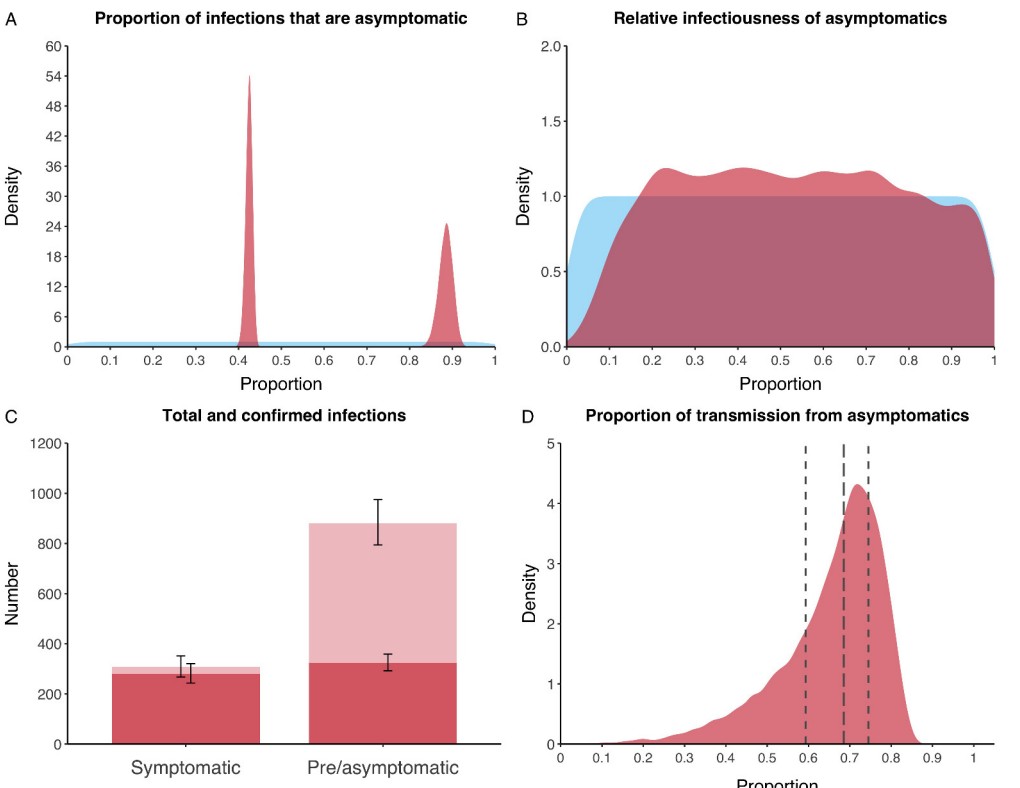

**Appendix 2—figure 21.** Proportion of infections that are asymptomatic and their contribution to transmission. (**A**) Prior (blue) and posterior (red) probability distribution for the proportion progressing to asymptomatic infections. The left hand peak is for passengers, whilst the right hand peak is for crew. (**B**) Prior (blue) and posterior (red) probability distribution for the relative infectiousness of asymptomatic infections. (**C**) Number of pre- and asymptomatic infections and symptomatic cases detected (dark red) and not detected (light red) during the outbreak. Error bars indicate 95% posterior intervals. (**D**) Posterior probability distribution for proportion of transmission that is from asymptomatic individuals. Dashed and dotted lines show median and interquartile range respectively.

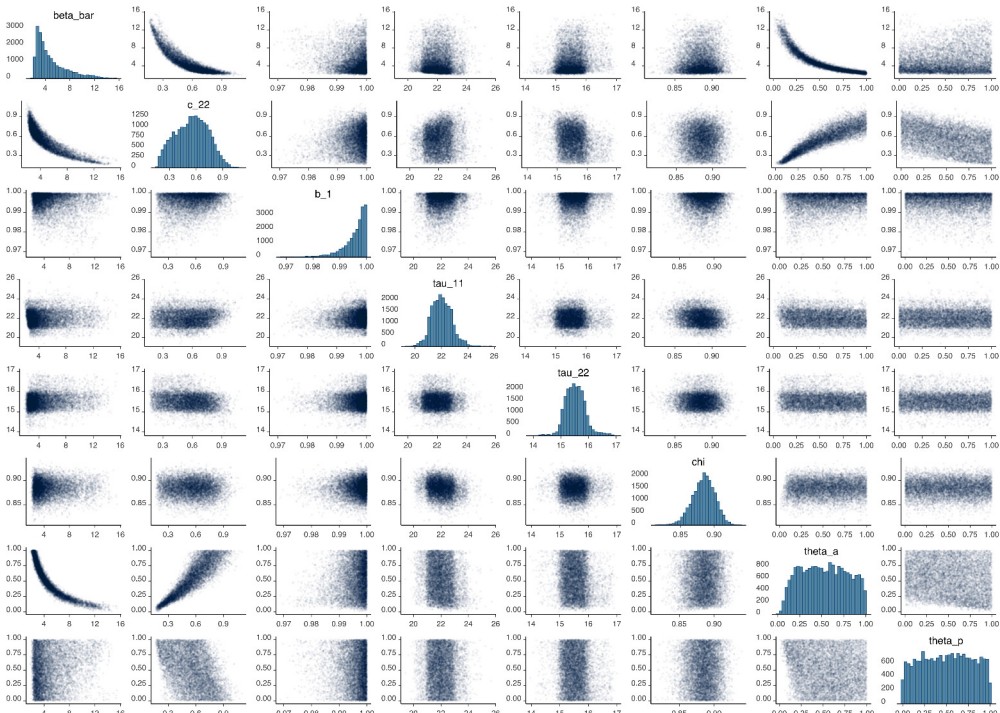

**Appendix 2—figure 22.** Parameter correlation plot containing parameter values from 10,000 samples of the joint posterior distribution found during MCMC model calibration. See *Table 2* in the main article for parameter definitions and descriptions.

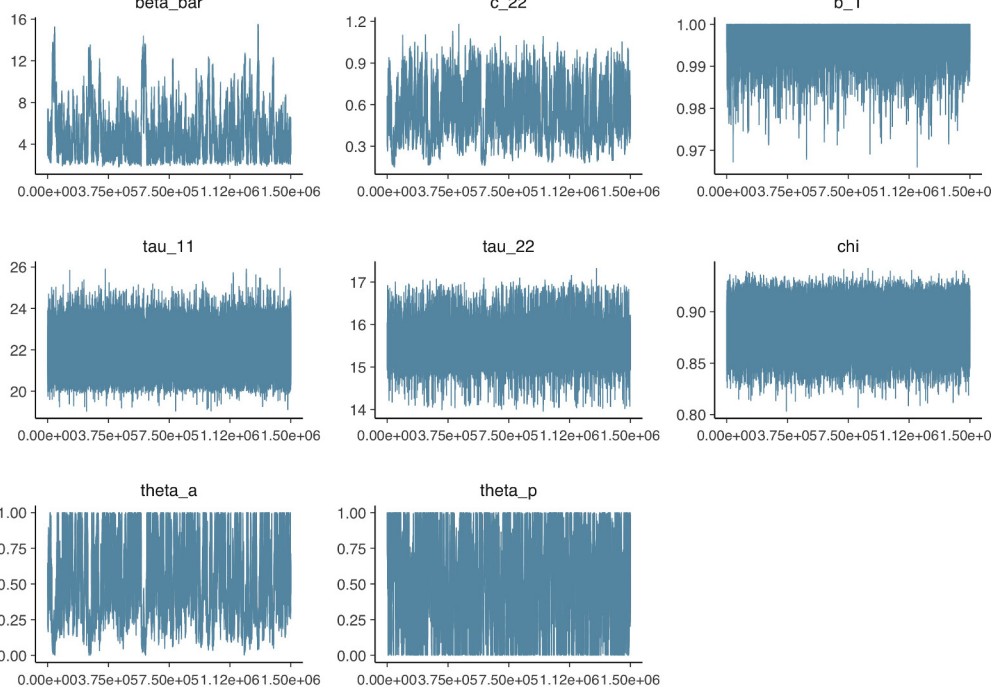

**Appendix 2—figure 23.** Parameter trace plot showing all 1.5 million samples from the MCMC model calibration sequentially. See *Table 2* in the main article for parameter definitions and descriptions.

## 7. Duration of latent period: 8.8 days

Assumes the average duration for the latent period is $1/v = 8.8$ days (*Jiang et al., 2020*), compared to 4.3 in the primary analysis.

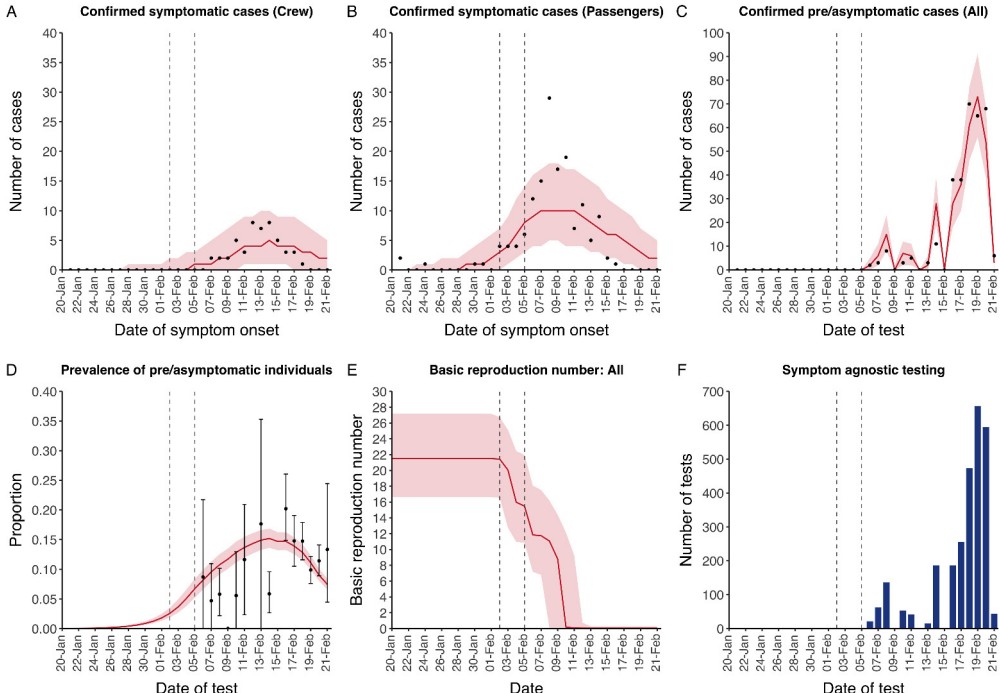

**Appendix 2—figure 24.** Data from the Diamond Princess and model calibration. Figure shows data from the Diamond Princess (points (**A–D**) and bars (**F**)) and results from model calibration. Red lines = median, shading = 95% posterior plus observational interval (**A–C**) and 95% posterior interval only (**D–E**). Two vertical lines show the date of the first confirmed diagnosis (left) and the start of quarantine measures (right). (**A–B**) show confirmed symptomatic cases among crew (**A**) and passengers (**B**) with a reported date of onset; (**C**) shows confirmed pre- or asymptomatic individuals by test date; (**D**) shows the prevalence of pre/asymptomatic individuals by test date. Points and error bars show point estimates and 95% confidence intervals; (**E**) shows the basic reproduction number over time for the ship as a whole, reflecting the drop in contact rates (**F**) shows the number of tests administered irrespective of symptoms, by test date.

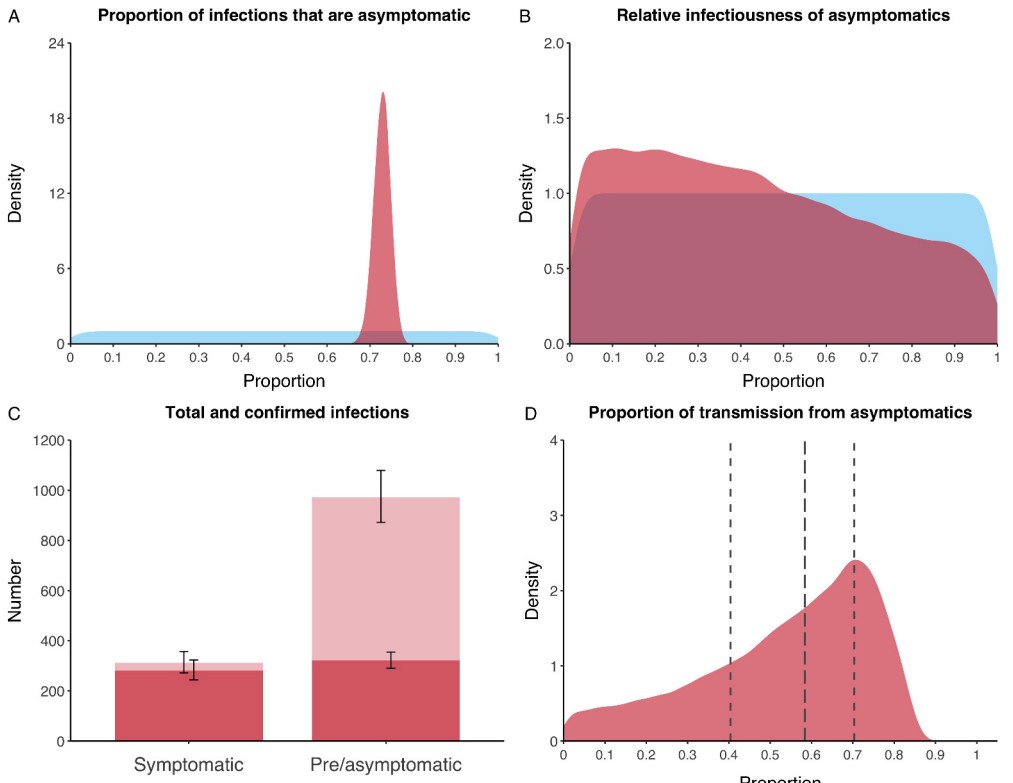

**Appendix 2—figure 25.** Proportion of infections that are asymptomatic and their contribution to transmission. (**A**) Prior (blue) and posterior (red) probability distribution for the proportion progressing to asymptomatic infections. (**B**) Prior (blue) and posterior (red) probability distribution for the relative infectiousness of asymptomatic infections. (**C**) Number of pre- and asymptomatic infections and symptomatic cases detected (dark red) and not detected (light red) during the outbreak. Error bars indicate 95% posterior intervals. (**D**) Posterior probability distribution for proportion of transmission that is from asymptomatic individuals. Dashed and dotted lines show median and interquartile range respectively.

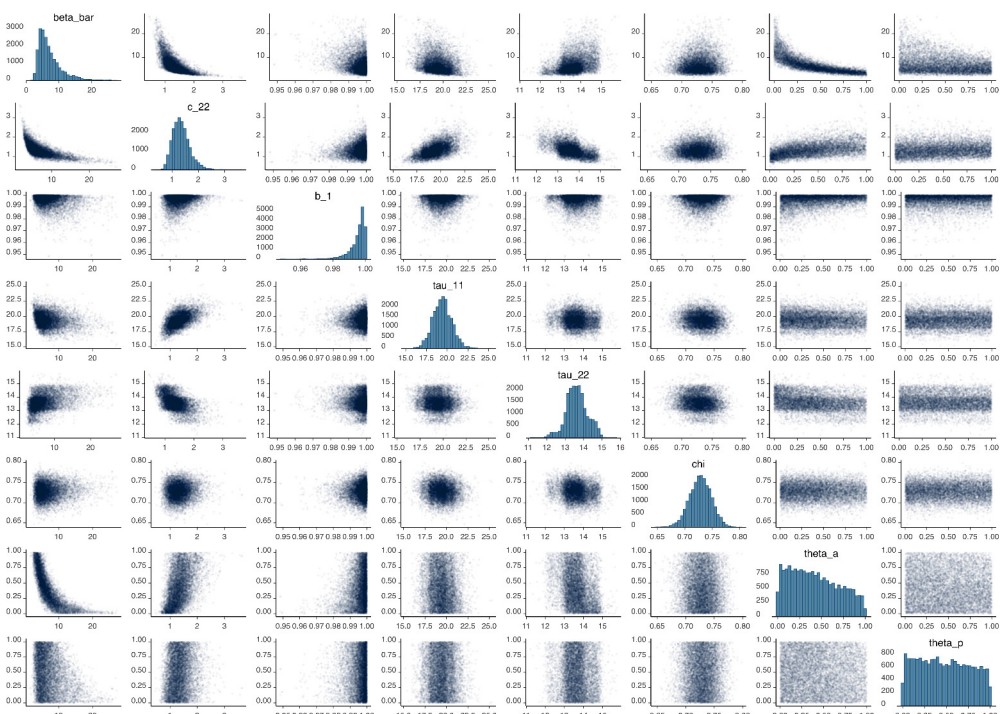

**Appendix 2—figure 26.** Parameter correlation plot containing parameter values from 10,000 samples of the joint posterior distribution found during MCMC model calibration. See *Table 2* in the main article for parameter definitions and descriptions.

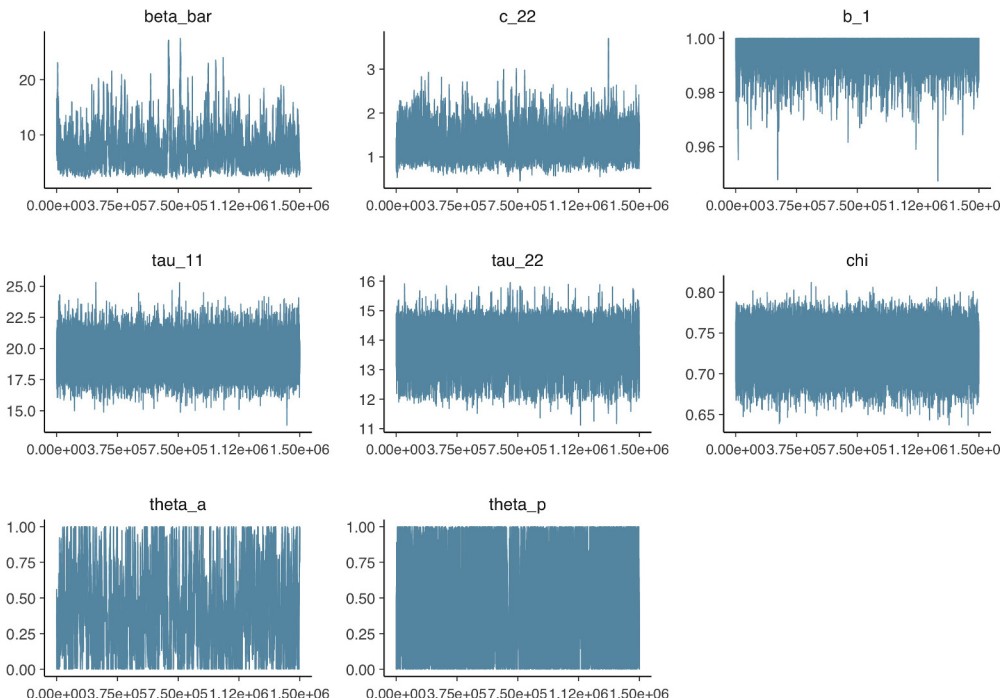

**Appendix 2—figure 27.** Parameter trace plot showing all 1.5 million samples from the MCMC model calibration sequentially. See *Table 2* in the main article for parameter definitions and descriptions.

## 8. Duration of asymptomatic infection: 2.5 days

Assumes the average duration of asymptomatic infection is $1/\gamma_a = 2.5$ days, compared to 5 days in the primary analysis.

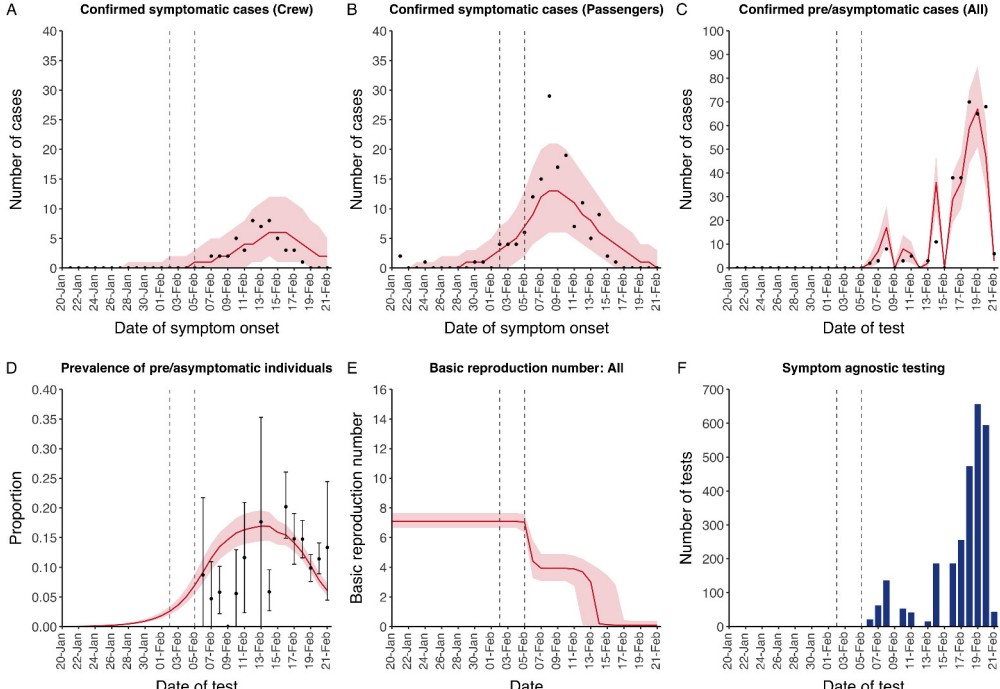

**Appendix 2—figure 28.** Data from the Diamond Princess and model calibration. Figure shows data from the Diamond Princess (points (**A–D**) and bars (**F**)) and results from model calibration. Red lines = median, shading = 95% posterior plus observational interval (**A–C**) and 95% posterior interval only (**D–E**). Two vertical lines show the date of the first confirmed diagnosis (left) and the start of quarantine measures (right). (**A–B**) show confirmed symptomatic cases among crew (**A**) and passengers (**B**) with a reported date of onset; (**C**) shows confirmed pre- or asymptomatic individuals by test date; (**D**) shows the prevalence of pre/asymptomatic individuals by test date. Points and error bars show point estimates and 95% confidence intervals; (**E**) shows the basic reproduction number over time for the ship as a whole, reflecting the drop in contact rates (**F**) shows the number of tests administered irrespective of symptoms, by test date.

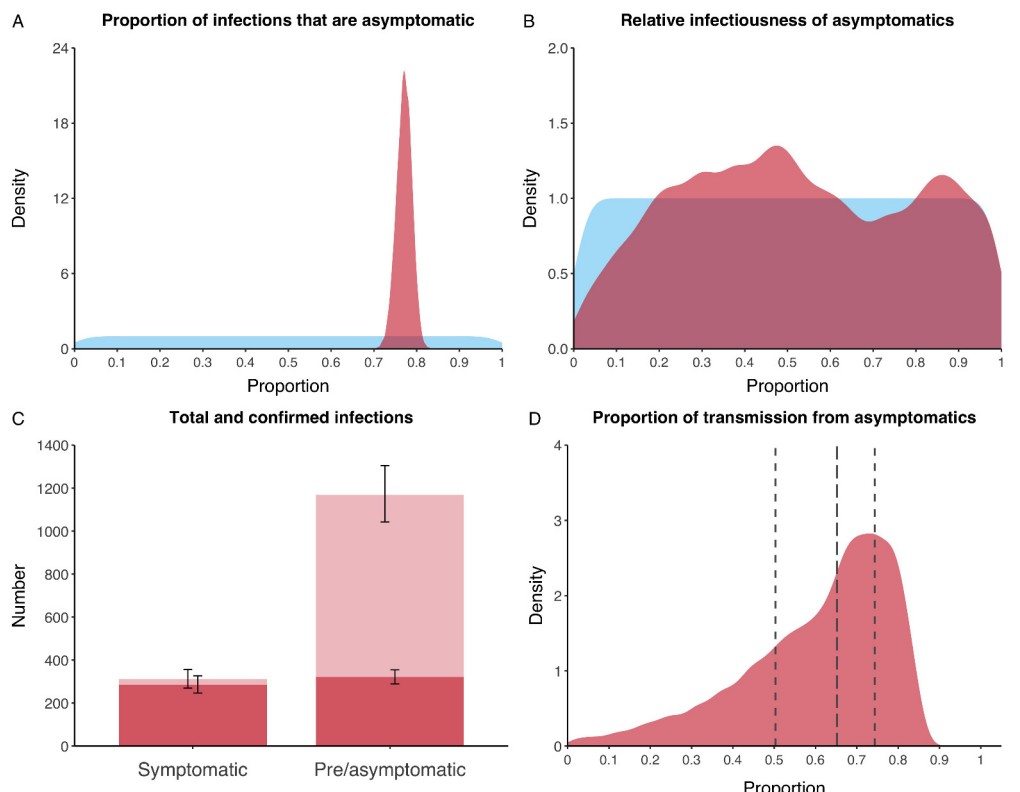

**Appendix 2—figure 29.** Proportion of infections that are asymptomatic and their contribution to transmission. (**A**) Prior (blue) and posterior (red) probability distribution for the proportion progressing to asymptomatic infections. (**B**) Prior (blue) and posterior (red) probability distribution for the relative infectiousness of asymptomatic infections. (**C**) Number of pre- and asymptomatic infections and symptomatic cases detected (dark red) and not detected (light red) during the outbreak. Error bars indicate 95% posterior intervals. (**D**) Posterior probability distribution for proportion of transmission that is from asymptomatic individuals. Dashed and dotted lines show median and interquartile range respectively.

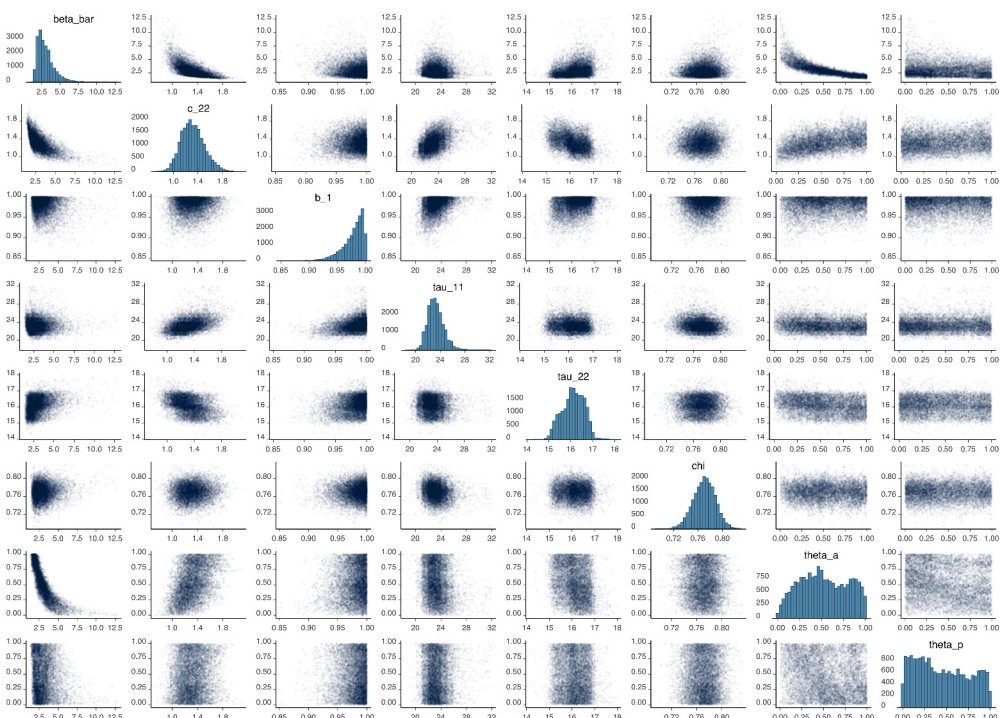

**Appendix 2—figure 30.** Parameter correlation plot containing parameter values from 10,000 samples of the joint posterior distribution found during MCMC model calibration. See *Table 2* in the main article for parameter definitions and descriptions.

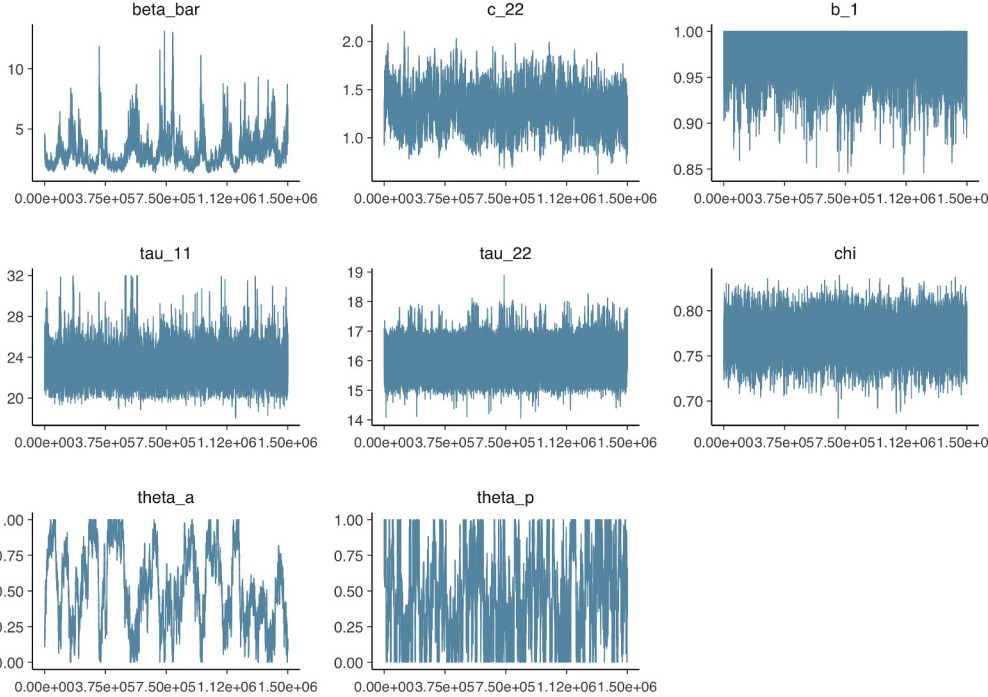

**Appendix 2—figure 31.** Parameter trace plot showing all 1.5 million samples from the MCMC model calibration sequentially. See *Table 2* in the main article for parameter definitions and descriptions.

## 9. Duration of asymptomatic infection: 10 days

Assumes the average duration of asymptomatic infection is $1/\gamma_a = 10$ days, compared to 5 days in the primary analysis.

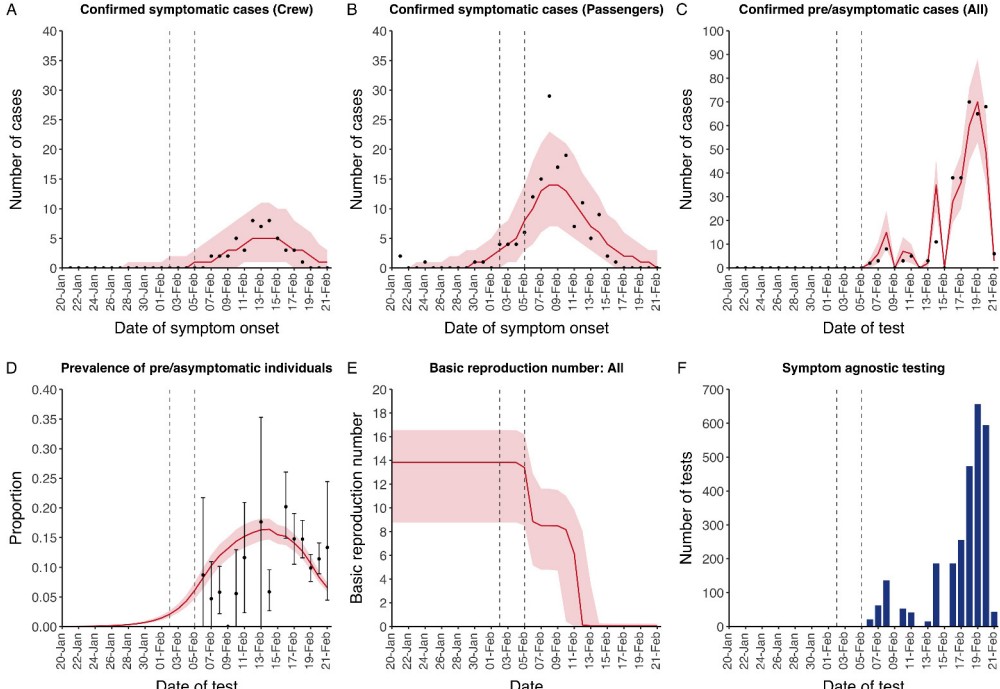

**Appendix 2—figure 32.** Data from the Diamond Princess and model calibration. Figure shows data from the Diamond Princess (points (**A–D**) and bars (**F**)) and results from model calibration. Red lines = median, shading = 95% posterior plus observational interval (**A–C**) and 95% posterior interval only (**D–E**). Two vertical lines show the date of the first confirmed diagnosis (left) and the start of quarantine measures (right). (**A–B**) show confirmed symptomatic cases among crew (**A**) and passengers (**B**) with a reported date of onset; (**C**) shows confirmed pre- or asymptomatic individuals by test date; (**D**) shows the prevalence of pre/asymptomatic individuals by test date. Points and error bars show point estimates and 95% confidence intervals; (**E**) shows the basic reproduction number over time for the ship as a whole, reflecting the drop in contact rates (**F**) shows the number of tests administered irrespective of symptoms, by test date.

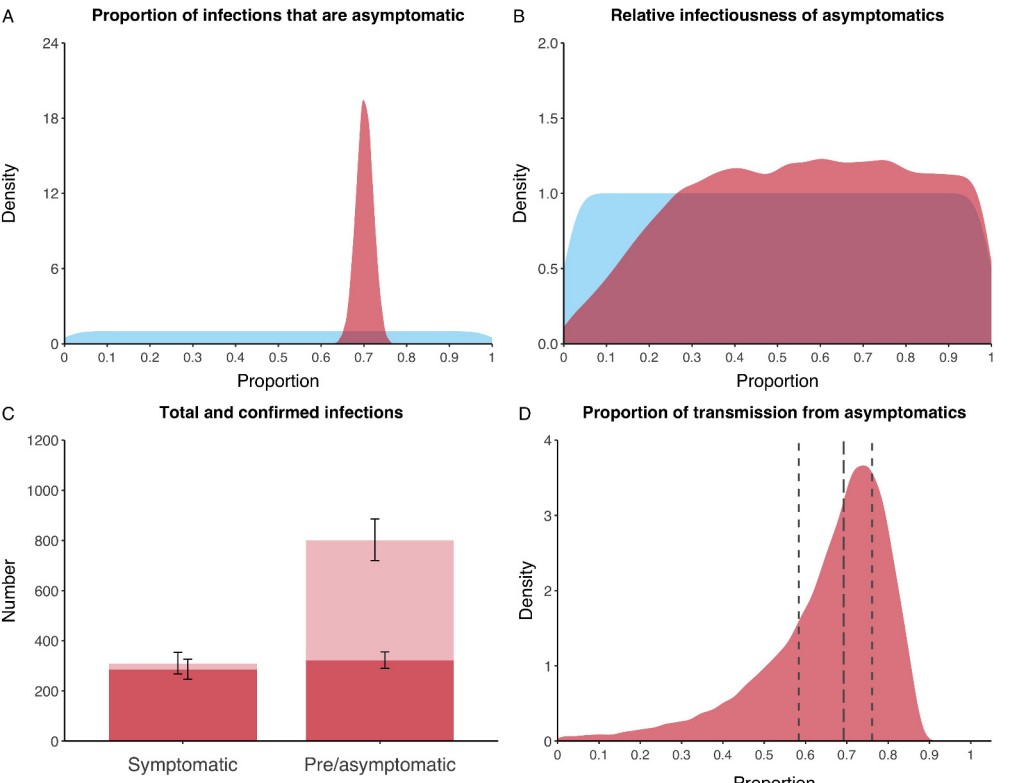

**Appendix 2—figure 33.** Proportion of infections that are asymptomatic and their contribution to transmission. (**A**) Prior (blue) and posterior (red) probability distribution for the proportion progressing to asymptomatic infections. (**B**) Prior (blue) and posterior (red) probability distribution for the relative infectiousness of asymptomatic infections. (**C**) Number of pre- and asymptomatic infections and symptomatic cases detected (dark red) and not detected (light red) during the outbreak. Error bars indicate 95% posterior intervals. (**D**) Posterior probability distribution for proportion of transmission that is from asymptomatic individuals. Dashed and dotted lines show median and interquartile range respectively.

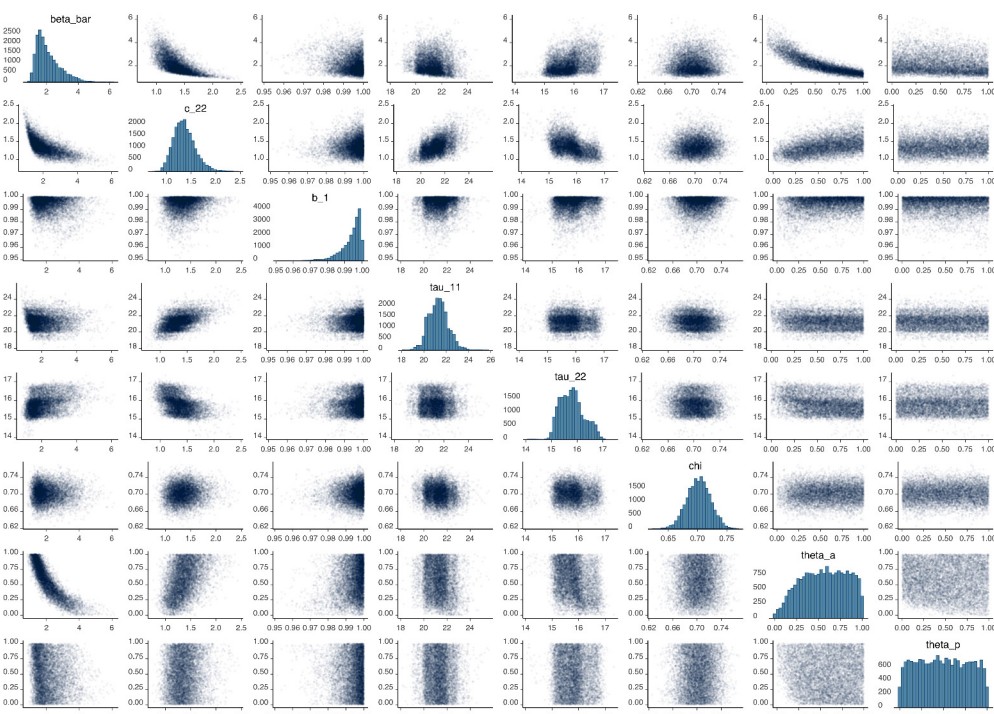

**Appendix 2—figure 34.** Parameter correlation plot containing parameter values from 10,000 samples of the joint posterior distribution found during MCMC model calibration. See *Table 2* in the main article for parameter definitions and descriptions.

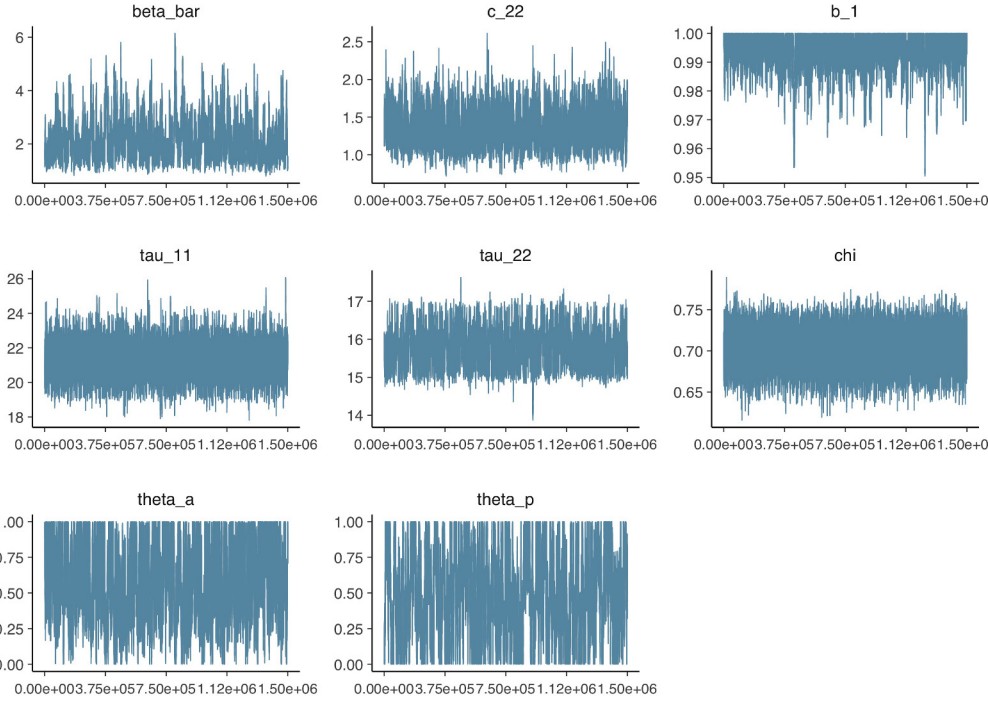

**Appendix 2—figure 35.** Parameter trace plot showing all 1.5 million samples from the MCMC model calibration sequentially. See *Table 2* in the main article for parameter definitions and descriptions.

## 10. Alternative distribution of n = 35 confirmed pre/asymptomatic cases

Assumes that n = 35 confirmed pre/asymptomatic cases without a test are apportioned to the last possible day (13th Feb), compared to proportional to the total number of tests administered over 6th-13th Feb in the primary analysis.

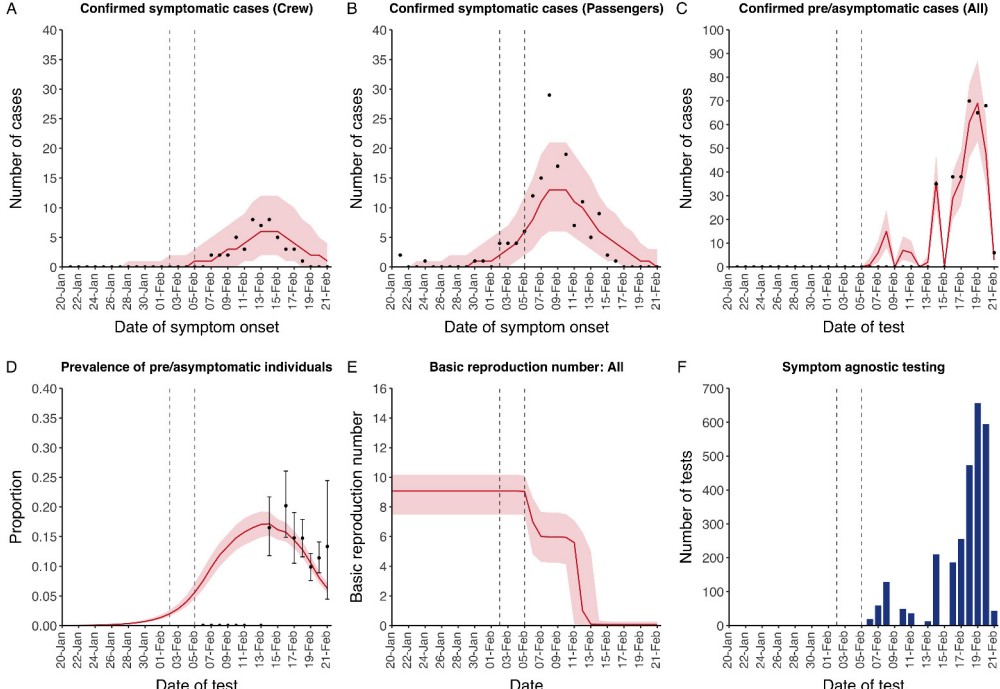

**Appendix 2—figure 36.** Data from the Diamond Princess and model calibration. Figure shows data from the Diamond Princess (points (**A–D**) and bars (**F**)) and results from model calibration. Red lines = median, shading = 95% posterior plus observational interval (**A–C**) and 95% posterior interval only (**D–E**). Two vertical lines show the date of the first confirmed diagnosis (left) and the start of quarantine measures (right). (**A–B**) show confirmed symptomatic cases among crew (**A**) and passengers (**B**) with a reported date of onset; (**C**) shows confirmed pre- or asymptomatic individuals by test date; (**D**) shows the prevalence of pre/asymptomatic individuals by test date. Points and error bars show point estimates and 95% confidence intervals; (**E**) shows the basic reproduction number over time for the ship as a whole, reflecting the drop in contact rates (**F**) shows the number of tests administered irrespective of symptoms, by test date.

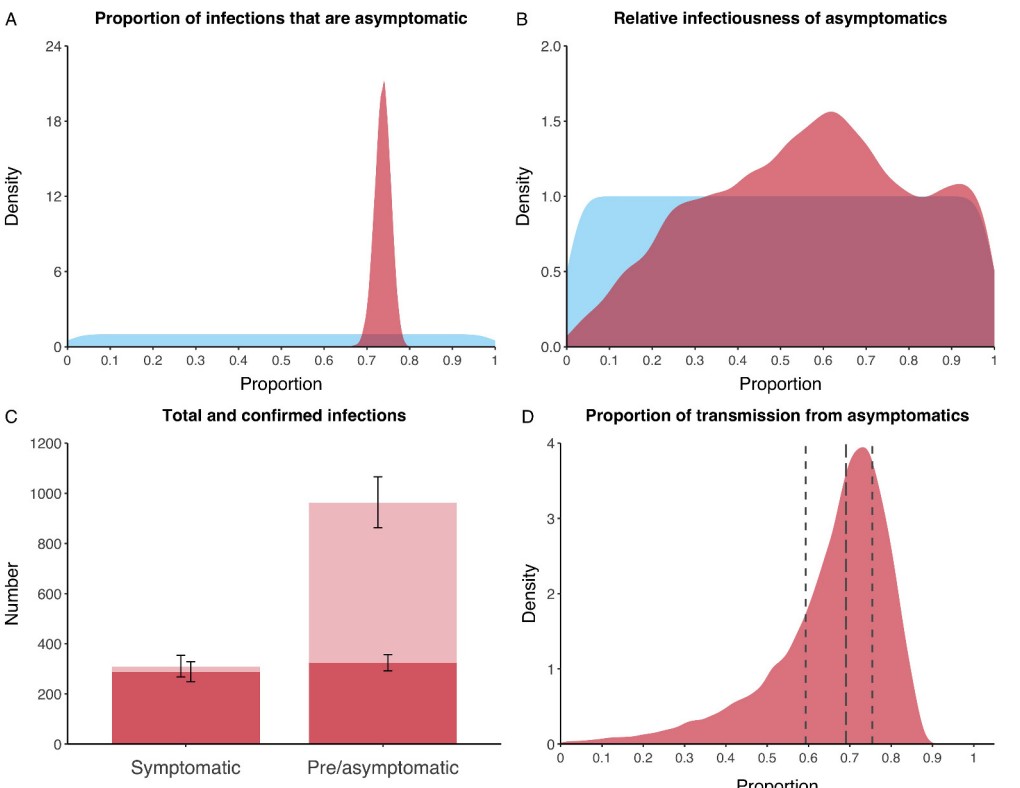

**Appendix 2—figure 37.** Proportion of infections that are asymptomatic and their contribution to transmission. (**A**) Prior (blue) and posterior (red) probability distribution for the proportion progressing to asymptomatic infections. (**B**) Prior (blue) and posterior (red) probability distribution for the relative infectiousness of asymptomatic infections. (**C**) Number of pre- and asymptomatic infections and symptomatic cases detected (dark red) and not detected (light red) during the outbreak. Error bars indicate 95% posterior intervals. (**D**) Posterior probability distribution for proportion of transmission that is from asymptomatic individuals. Dashed and dotted lines show median and interquartile range respectively.

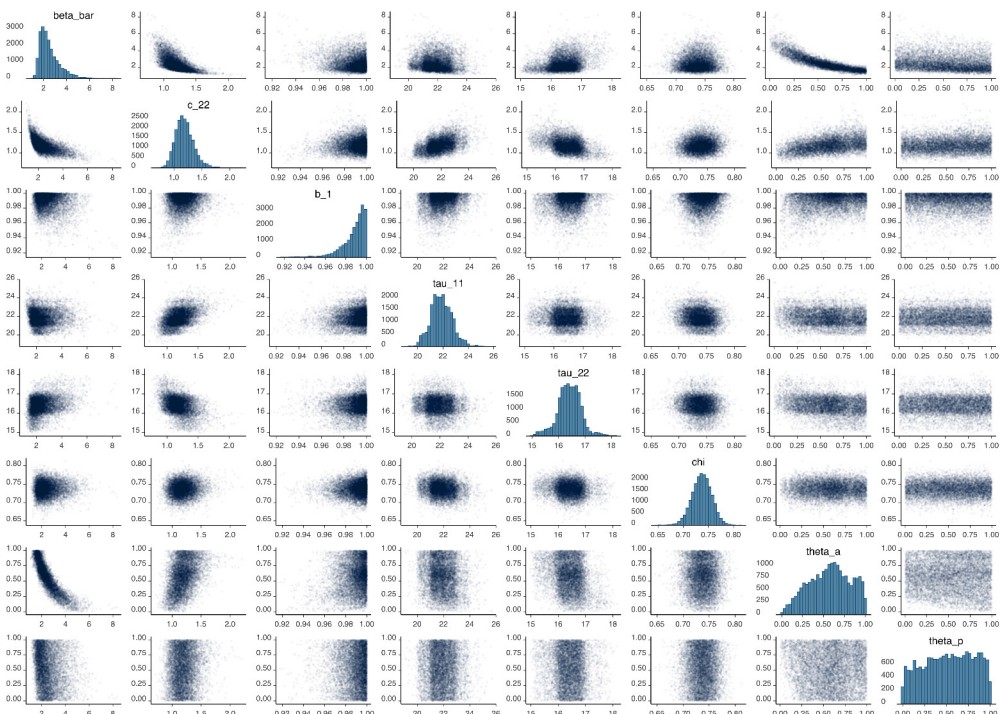

**Appendix 2—figure 38.** Parameter correlation plot containing parameter values from 10,000 samples of the joint posterior distribution found during MCMC model calibration. See *Table 2* in the main article for parameter definitions and descriptions.

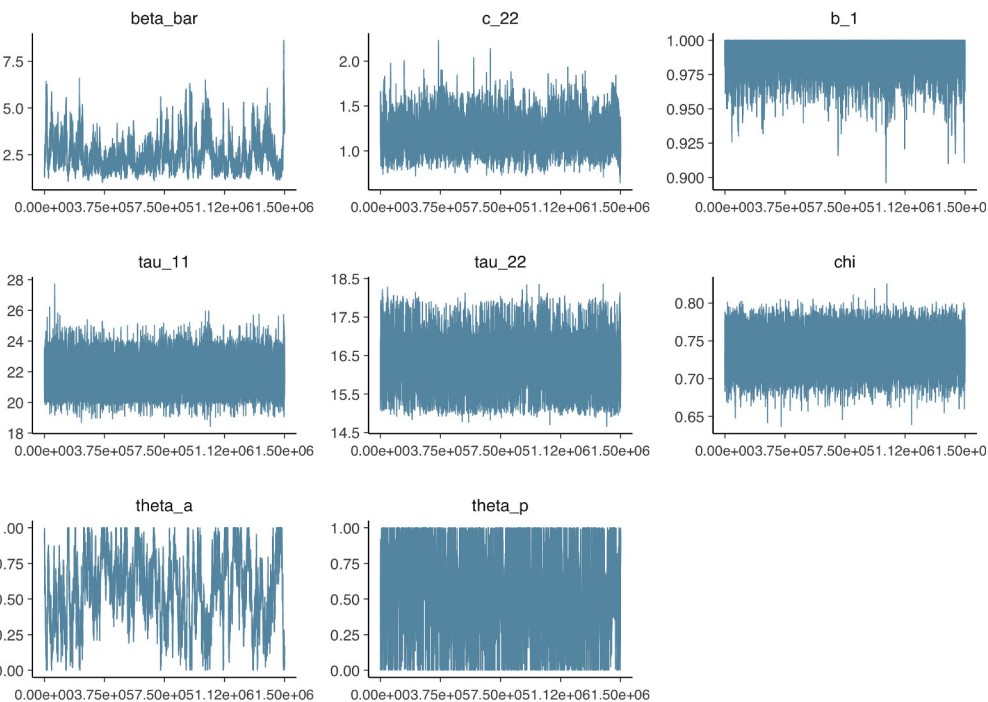

**Appendix 2—figure 39.** Parameter trace plot showing all 1.5 million samples from the MCMC model calibration sequentially. See *Table 2* in the main article for parameter definitions and descriptions.

