## [Decision Letter]

**Acceptance summary:**

This is a very detailed model-based workup of the Diamond Princess outbreak that is thoughtfully done, extracts tremendous insight from a messy dataset, and provides a good example of how rich such outbreak workups can be in elucidating features of transmission.

**Decision letter after peer review:**

Thank you for submitting your article "The contribution of asymptomatic SARS-CoV-2 to transmission – a modelling analysis of the Diamond Princess outbreak" for consideration by *eLife*. Your article has been reviewed by two peer reviewers, including Marc Lipsitch as the Reviewing Editor and Reviewer #1, and the evaluation has been overseen by Senior Editor.

The reviewers have discussed the reviews with one another and the Reviewing Editor has drafted this decision to help you prepare a revised submission.

We would like to draw your attention to changes in our revision policy that we have made in response to COVID-19 (https://elifesciences.org/articles/57162). Specifically, we are asking editors to accept without delay manuscripts, like yours, that they judge can stand as *eLife* papers without additional data, even if they feel that they would make the manuscript stronger. Thus the revisions requested below mainly address clarity and presentation (one sensitivity analysis requested in point 7 and one optionally in point 6).

We thank the authors for their patience during the review process, during which expert reviewers have been understandably rather overstretched.

Summary:

Excellent paper providing careful analysis of valuable data on asymptomatic transmission. [From reviewer 3]: This article is a very important analysis that provides a result of immediate interest to the field as well as to policymakers dealing with the pandemic. The analysis is conducted well and the manuscript describes the approach well, as well as providing several sensitivity analyses to demonstrate the robustness of the results. I have a few suggestions for pieces of the analysis to clarify and potential sensitivity analyses to add, but I believe this work should be published soon as it is important information to contribute to the discussion in the field.

Essential revisions:

1) I wish some more intuition (or any) could be provided for why infectiousness of asymptomatics is not identified but proportion asymptomatic so strongly is and contribution to transmission somewhat is.

2) I wonder how sensitive the results are to the assumptions about the flat contagiousness over the natural history. If symptomatics were most infectious during their early days of symptoms (as viral load data suggest) then their removal would do less to reduce their infectiousness and they would contribute more to transmission (I think). Some sensitivity analysis to this would be helpful.

3) How well supported by data is the 1 day mean (and exponential distribution, which is probably wrong) of time to removal post-test+? Does changing this to an immediate end to infectiousness (presumably people are very careful around known positives) change things?

4) The conclusions are critically conditional on the setting. This needs to be emphasized in a whole paragraph of the Discussion, including how they might change in a setting of less intense testing and removal. For example if symptomatic test+ people are removed very fast (on average) this will increase the proportion of transmission by asymptomatics, at least in a period where only symptomatics are tested. So it is important to avoid the situation where this high proportion of contribution to transmission becomes interpreted as a constant of nature rather than a property of transmission on the DP.

5) I would like to see more explanation of why it is possible to id the % asymptomatic, and their contribution to transmission, but not a quantity that is related to the ratio of these: their relative infectiousness.

6) It appears that asymptomatic infections occur independently (i.e., an individual infected by someone who is asymptomatic is no more or less likely to be asymptomatic themselves). Is this correct? If so, it is worth mentioning the assumption and possible consequences of it in the Materials and methods and Discussion. This could be an additional sensitivity analysis as well.

7) It would be helpful to explain the assumptions about the testing regime in the main text. Sensitivity analyses should also consider if the testing is not truly random sampling of non-symptomatic individuals. E.g., what if individuals with infection are more likely to get tested (low-level symptoms they do not report)? Or what if individuals with infection are less likely to get tested (health-seeking behavior)? Since it relates directly to the number of asymptomatic infections in the model, it might have a larger effect than some of the other sensitivity tests.

8) If possible, it would be good to see the percentage of infections that are by pre and asymptomatic individuals by date, to see how it changes with the progression of the epidemic and the progression to symptoms within individuals.

---

## [Author Response]

Essential revisions:1) I wish some more intuition (or any) could be provided for why infectiousness of asymptomatics is not identified but proportion asymptomatic so strongly is and contribution to transmission somewhat is.

We have added additional details to the Results section under ‘Asymptomatic infections’ to help provide more intuition for our results.

In particular, we have noted that with respect to the proportion asymptomatic:

“The strong identifiability of this parameter is driven by the relative proportions of individuals testing positive with and without symptoms, combined with the time delay between symptom-based and symptom-agnostic testing”.

Thereafter, with respect to the unidentifiability of relative infectiousness of asymptomatics, we note that:

“This is because the relative infectiousness of asymptomatic infections was degenerate with the overall contact rate, meaning the data was consistent with either relatively frequent contact with less infectious individuals or relatively infrequent contact with more infectious individuals (see Figure 1—figure supplement 1).”

Finally, with respect to the contribution of asymptomatics to overall transmission, we detail that:

“The reason this estimate is not effectively 0-100%, as might be expected by the unidentifiable relative infectiousness, is the combination of the strongly identified, relatively high proportion of infections that are asymptomatic and the non-linear relationship between the relative infectiousness of asymptomatics and their contribution to transmission, which quickly saturates to its maximal value (see Figure 2—figure supplement 1). The result is that only a modest relative infectiousness is required to produce a non-trivial contribution to transmission.”

2) I wonder how sensitive the results are to the assumptions about the flat contagiousness over the natural history. If symptomatics were most infectious during their early days of symptoms (as viral load data suggest) then their removal would do less to reduce their infectiousness and they would contribute more to transmission (I think). Some sensitivity analysis to this would be helpful.

We thank the reviewer for this interesting question. We have added the following paragraph to the Discussion, under ‘Limitations’, that considers both this point as well as point 3 below:

“Our model also assumed that the infectiousness of all transmissible states was constant over time. If instead symptomatic individuals are most infectious immediately after symptom onset (He et al., 2020), an estimate of their contribution to transmission would in principle increase. Given the likely heightened awareness of symptoms on board however, such an increase is likely to be marginal. Indeed, since our assumption of an average one-day, exponentially distributed delay between symptom onset and removal from the ship is likely to be an overestimate, a prompter removal distribution would at least in part offset such an increase.”

3) How well supported by data is the 1 day mean (and exponential distribution, which is probably wrong) of time to removal post-test+? Does changing this to an immediate end to infectiousness (presumably people are very careful around known positives) change things?

We have addressed this point together with point 2 in a paragraph added to the Discussion, under ‘Limitations’. See point 2 above for details.

4) The conclusions are critically conditional on the setting. This needs to be emphasized in a whole paragraph of the Discussion, including how they might change in a setting of less intense testing and removal. For example if symptomatic test+ people are removed very fast (on average) this will increase the proportion of transmission by asymptomatics, at least in a period where only symptomatics are tested. So it is important to avoid the situation where this high proportion of contribution to transmission becomes interpreted as a constant of nature rather than a property of transmission on the DP.

We agree that this is an important point. We have added a paragraph to the Discussion under ‘Interpretation’ to highlight this:

“It is important to note that our conclusion that asymptomatic infections may have contributed substantially to ongoing transmission is critically dependent on the setting. In this case symptomatic infections were quickly identified and removed from the ship before symptom-agnostic testing began, thereby leaving asymptomatic infections to dominate transmission. Such dominance should therefore not be interpreted as a constant of nature, but instead an important consideration in settings where prompt isolation of symptomatic infections is already in place but with little to no consideration for asymptomatic infections.”

5) I would like to see more explanation of why it is possible to id the % asymptomatic, and their contribution to transmission, but not a quantity that is related to the ratio of these: their relative infectiousness.

We have added additional details to the Results section to help provide more intuition for our results. See point 1 above for details.

6) It appears that asymptomatic infections occur independently (i.e., an individual infected by someone who is asymptomatic is no more or less likely to be asymptomatic themselves). Is this correct? If so, it is worth mentioning the assumption and possible consequences of it in the Materials and methods and Discussion. This could be an additional sensitivity analysis as well.

The reviewer is correct. We have added a sentence to the Materials and methods section under ‘Model’ to clarify this assumption:

“This proportion equates to a universal probability of becoming either presymptomatic or asymptomatic, independent of who infected whom.”

We have also added a paragraph to the Discussion under ‘Limitations’ to highlight this assumption:

“A similar simplification was made by assuming that the probability of an individual progressing to either a presymptomatic or asymptomatic infection was independent of who infected whom. It is possible, however, that transmission from a symptomatic infection may be more likely to ultimately result in another symptomatic infection, owing to a higher infecting dose for example.”

7) It would be helpful to explain the assumptions about the testing regime in the main text. Sensitivity analyses should also consider if the testing is not truly random sampling of non-symptomatic individuals. E.g., what if individuals with infection are more likely to get tested (low-level symptoms they do not report)? Or what if individuals with infection are less likely to get tested (health-seeking behavior)? Since it relates directly to the number of asymptomatic infections in the model, it might have a larger effect than some of the other sensitivity tests.

We agree with the reviewer that this is an important consideration. Firstly, we have added a paragraph to the Materials and methods section under ‘Model’ making clearer our assumptions regarding testing:

“Symptom-agnostic testing was assumed to have been random amongst those not reporting symptoms and no delay was introduced between testing and removal of those that tested positive from the ship. […] All testing was assumed to have 100% sensitivity and specificity.”

Secondly, we have added sensitivity analyses considering biased symptom-agnostic testing. The scenarios have been added the Materials and methods section under ‘Sensitivity analyses’:

“Thirdly, we considered the impact of biased symptom-agnostic testing. Specifically, we first assumed that those that would test positive were 50% more likely to be tested, before then assuming that those that would test negative were 50% more likely to be tested, both compared to purely random testing as per the primary analysis.”

The results of these analyses are presented in Appendix 2—figures 12-19. A new paragraph summarising these results has also been added to the Results section, under ‘Sensitivity analyses’:

“Biased symptom-agnostic testing had a greater impact on the results. […] Conversely, those testing positive having a 50% higher probability of being tested compared to the primary analysis led to a corresponding smaller proportion of infections that were asymptomatic (66%, 61-71%), total number infected (1,051, 965-1,161) and contribution of asymptomatics to transmission (59%, 9-79%) (see Appendix 2—figures 16-19).”

Finally, we have added some discussion of biased symptom-agnostic testing to the Discussion, under ‘Limitations’:

“Whilst symptom-agnostic testing provided valuable insights into the pre- and asymptomatic states, such testing was not necessarily random, as was assumed in our primary analysis. Indeed, it is known that individuals were generally screened in reverse age order (Field Briefing: Diamond Princess COVID-19 Cases, 2020). Sensitivity analyses considering biased testing still produced non-trivial results for the proportion of infections that were asymptomatic however.”

8) If possible, it would be good to see the percentage of infections that are by pre and asymptomatic individuals by date, to see how it changes with the progression of the epidemic and the progression to symptoms within individuals.

Figure 3 has been added, showing these results. A reference to Figure 3 has been added to the Results section under ‘Asymptomatic infections’:

“Figure 3 shows the instantaneous proportion of transmission from symptomatic (A), presymptomatic (B) and asymptomatic (C) individuals over the course of the epidemic.”

Description of how the contents of Figure 3 were calculated has been added to Appendix 1, under ‘Model outputs’:

“The instantaneous proportion of transmission from either symptomatic, presymptomatic or asymptomatic individuals was calculated by dividing the number of infections generated by the respective infected state in the previous timestep by the total number of new infections in the previous timestep.”